# Overcoming the coherence time barrier in quantum machine learning on temporal data

Fangjun Hu [1,4], Saeed A. Khan [1,4], Nicholas T. Bronn [2], Gerasimos Angelatos [1,3], Graham E. Rowlands [3], Guilhem J. Ribeill[3] & Hakan E. Türeci[1] ✉

The practical implementation of many quantum algorithms known today is limited by the coherence time of the executing quantum hardware and quantum sampling noise. Here we present a machine learning algorithm, NISQRC, for qubit-based quantum systems that enables inference on temporal data over durations unconstrained by decoherence. NISQRC leverages mid-circuit measurements and deterministic reset operations to reduce circuit executions, while still maintaining an appropriate length persistent temporal memory in the quantum system, confirmed through the proposed Volterra Series analysis. This enables NISQRC to overcome not only limitations imposed by finite coherence, but also information scrambling in monitored circuits and sampling noise, problems that persist even in hypothetical fault-tolerant quantum computers that have yet to be realized. To validate our approach, we consider the channel equalization task to recover test signal symbols that are subject to a distorting channel. Through simulations and experiments on a 7-qubit quantum processor we demonstrate that NISQRC can recover arbitrarily long test signals, not limited by coherence time.

The development of machine learning algorithms that can handle data with temporal or sequential dependencies, such as recurrent neural networks[1] and transformers[2], has revolutionized fields like natural language processing[3]. Real-time processing of streaming data, also known as online inference, is essential for applications such as edge computing, control[4], and forecasting[5]. The use of physical systems whose evolution naturally entails temporal correlations appears, at first sight, to be ideally suited for such applications. An emerging approach to learning, referred to as physical neural networks (PNNs)[6–9], employs a wide variety of physical systems to compute a trainable transformation on an input signal. A branch of PNNs that has proven well suited to online data processing is physical reservoir computing[10], distinguished by its trainable component being only a linear projector acting on the observable state of the physical system[11]. This approach has the enormous benefit of fast convex optimization through singular value decomposition routines

and has already enabled temporal learning on various hardware platforms[4,12–15].

Among many physical systems considered for PNNs, quantum systems are believed to offer an enormous potential for more scalable, resource-efficient, and faster machine learning[16–23], due to their evolution taking place in the Hilbert space that scales exponentially with the number of nodes[24–30]. However, quantum machine learning (QML) on present-day noisy intermediate-scale quantum (NISQ) hardware has so far been restricted to training and inference on low-dimensional static data due to several difficulties. A fundamental restriction is Quantum Sampling Noise (QSN) – the unavoidable uncertainty arising from the finite sampling of a quantum system – which limits the accuracy of both QML training and inference[9,31,32] even on fault-tolerant hardware. In addition, the optimization landscape for training quantum systems often features "barren plateaus"[33,34], which are regions where optimization becomes exponentially difficult. These

[1]Department of Electrical and Computer Engineering, Princeton University, Princeton, NJ, USA. [2]IBM Quantum, IBM T.J. Watson Research Center, Yorktown Heights, NY, USA. [3]RTX BBN Technologies, Cambridge, MA, USA. [4]These authors contributed equally: Fangjun Hu, Saeed A. Khan. ✉e-mail: tureci@princeton.edu

plateaus, especially in the presence of QSN, present a significant challenge to implementing QML at scales relevant to practical applications.

Two further concerns arise when considering inference on long data streams, which call into question whether quantum systems can even *in principle* be employed for online learning on streaming data. Firstly, without quantum error correction, the operation fidelities and finite coherence times of constituent quantum nodes place a limit on the size of data on which inference can be performed[35,36], which would appear to rule out inference on long data streams. Secondly, the nature of measurement on quantum systems imposes a fundamental constraint on continuous information extraction over long times. Backaction due to repeated measurements on quantum systems necessitated by inference on streaming data is expected to lead to the rapid distribution of information between different parts of the system, a phenomenon known as information scrambling and thermalization[37,38], making it extremely difficult to track or retrieve the information correlations in the input data. This constraint persists even in an ideal system with perfect coherence, such as one that may be realized by a fault-tolerant quantum computer. It is not known precisely what conditions must be satisfied to avoid information scrambling. For classical dynamical systems, a strict condition known as the *fading memory property*[39,40] is required for a physical system to retain a persistent temporal memory that does not degrade on indefinitely long data streams. This imposes restrictions on the design of a classical reservoir and in particular, how input data is encoded. Here, a mathematical framework known as Volterra Series theory[41] provides the basis for analyzing the memory properties of a classical dynamical system. Such a general theory for quantum systems has remained elusive so far.

Here we present a Volterra theory for quantum systems that accounts for measurement backaction, necessary for analyzing the conditions required to endow a quantum system with a persistent temporal memory on streaming data. Based on this Quantum Volterra Theory we propose an algorithm, NISQ Reservoir Computing (NISQRC), that leverages recent technical advances in mid-circuit measurements to process signals of arbitrary duration, not limited by the coherence time of constituent physical qubits (see Fig. 1). The property that enables inference on an indefinitely-long input signal – the ability to avoid measurement-induced thermalization at long times under repeated measurements due to a deterministic reset protocol – is intrinsic to the algorithm: it survives even in the presence of QSN, and does not require operating in a precisely-defined parameter subspace – and is thus unencumbered by barren plateaus.

Here, we demonstrate the practical viability of NISQRC through application to a task of technological relevance for communication systems, namely, the equalization of a wireless communication channel. Channel equalization aims to reconstruct a message streamed through a noisy, non-linear and distorting communication channel and has been employed in benchmarking reservoir computing architectures[11,14] as well as other machine learning algorithms[42,43]. This task poses a challenge for parametric circuit learning-based algorithms[19] because the number of symbols in the message, $N_{ts}$, to recover in the inference stage directly determines the length of the encoding circuit, which, in turn, is limited by the coherence time of the system. A more critical issue is that the recovery has to be done online, as the message is streamed, which structurally is not suitable for static encoding schemes. We demonstrate through numerical simulation (Results' subsection "Practical machine learning using temporal data") and experiments on a 7-qubit quantum processor (Results' subsection "Experimental results on the quantum system") that NISQRC enables quantum systems to process signals of arbitrary duration. Most significantly, this ability to continuously extract useful information from a single quantum circuit is not limited by coherence time. Instead, the quantum system's coherence influences the resulting memory time-scale; we show that by balancing the length of individual input

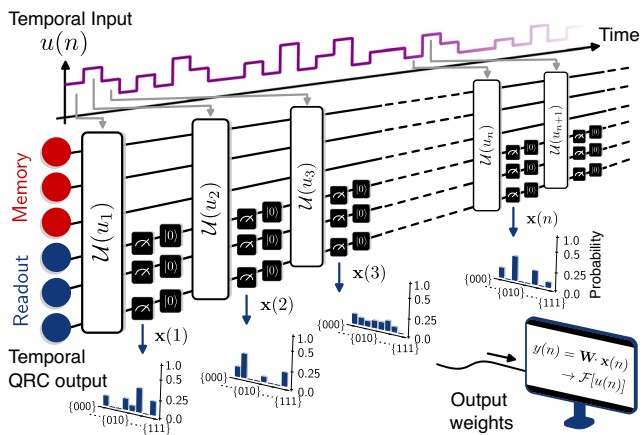

**Fig. 1 | Schematic representation of NISQRC architecture for machine learning on temporal data using a convex optimization algorithm on finitely-sampled partial measurements.** For concreteness, the architecture is shown for a quantum circuit with a projective computational basis readout; both the underlying quantum system and the measurement scheme can be much more general. Temporal input data is encoded into the evolution of the reservoir at every time-step $n$ via a quantum channel $\mathcal{U}(u_n)$; a non-trivial I/O map is enabled via partial readout and subsequent reset of a readout subsystem while a memory subsystem retains the memory of past inputs. Temporal quantum reservoir computing (QRC) output $\mathbf{x}(n)$ are obtained via measurements (more precisely, stochastic unbiased estimators $\bar{\mathbf{X}}(n)$ of expected features are constructed from $S$ repetitions of the experiment, see Method III A), and a learned linear combination is used to approximate the target functional $y(n)$ of $u_n$. The overall execution time of the circuit is $O(NS)$, where $N$ is the length of the input temporal sequence.

encoding steps with the rate of information extraction through mid-circuit measurements, it is possible to endow the circuit with a memory that is appropriate for the ML task at hand. Even in the limit of infinite coherence, temporal memory is still limited by this fundamental trade-off. Reliable inference on a time-dependent signal of duration $T_{run} = 117\,\mu s$ is demonstrated on a 7-qubit quantum processor with qubit lifetimes in the range $63\,\mu s - 164\,\mu s$ and $T_2 = 9\,\mu s - 231\,\mu s$. In our experiments, longer durations are restricted by limitations on mid-circuit buffer clearance. To leave no doubt that a persistent memory can be generated, we first compare the experimental results to numerical simulations with the same parameters, showing excellent agreement. Building on the accuracy of numerical simulations in the presence of finite coherence and our noise model, we explicitly demonstrate successful inference on a 5000 symbol signal: the resulting circuit duration is 500 times that of the individual qubit lifetimes.

Here, we also develop a method to efficiently sample from deep circuits under partial measurements. Simulating individual quantum trajectories for circuits with repeated measurements requires the traversal of ever-branching paths conditioned on the measurement results, which becomes rapidly unfeasible for deep circuits. Our numerical method (see "Methods" subsection "The quantum Volterra theory and analysis of NISQRC") allows us to sample from repeated partial measurements on circuits of arbitrary depth. We use our scheme to numerically explore other seemingly reasonable encoding methods adopted in previous studies, showing that these can lead to a sharp decline in performance when the effect of measurement is properly accounted for. Drawing upon the Quantum Volterra Theory, we unveil the underlying cause: the absence of a persistent memory mechanism.

## Results
### Time-series processing in quantum systems
The general aim of computation on temporal data is expressed most naturally in terms of functionals of a time-dependent input

$\mathbf{u} = \{u_{-\infty}, \cdots, u_{-1}, u_0, u_1, \cdots, u_{\infty}\}$. A functional $\mathcal{F} : \mathbf{u} \mapsto \mathbf{y}$ maps a bounded function $\mathbf{u}$ to another arbitrary bounded function $\mathbf{y}$, where $\mathbf{y} = \{y_{-\infty}, \cdots, y_{-1}, y_0, y_1, \cdots, y_{\infty}\}$. Without loss of generality, these functions can be normalized; we choose $u_n \in [-1, 1]$ and $y_n \in [-1, 1]$. Within the reservoir computing paradigm[44], this processing is achieved by extracting outputs $\mathbf{x}(n)$, where $n$ is a temporal index, from a physical system evolving under said time-dependent stimulus $u_n \equiv \mathbf{u}(n)$. Learning then entails finding a set of optimal time-independent weights $\mathbf{w}$ to best approximate a desired $\mathcal{F}$ with a linear projector $y_n \equiv \mathbf{y}(n) = \mathbf{w} \cdot \mathbf{x}(n)$. If the physical system is sufficiently complex, its temporal response $\mathbf{x}(n)$ to a time-dependent stimulus $\mathbf{u}$ is universal in that it can be used to approximate a large set of functionals $\mathcal{F}[\mathbf{u}]$ with an error scaling inversely in system size and using only this simple linear output layer[27,28,45].

To analyze the utility of this learning framework, it proves useful to quantify the space of functionals $\mathcal{F}[\mathbf{u}]$ that are accessible. For classical non-linear systems, a firmly established means of doing so is a Volterra series representation of the input-output (I/O) map[39]:

$$x_j(n) = \sum_{k=0}^{\infty} \sum_{n_1=0}^{\infty} \cdots \sum_{n_k=n_{k-1}}^{\infty} h_k^{(j)}(n_1, \cdots, n_k) \prod_{\kappa=1}^{k} u_{n-n_\kappa} \tag{1}$$

where the Volterra kernels $h_k^{(j)}(n_1, \cdots, n_k)$ characterize the dependence of the systems' measured output features at time $n$ on its past inputs $u_{n-n_\kappa}$. Hence the support of $h_k^{(j)}$ over the the temporal domain $(n_1, \cdots, n_k)$ quantifies the notion of *memory* of a particular physical system, with the kernel order $k$ being the corresponding degree of nonlinearity of the map. Most importantly, the Volterra series representation describes a time-invariant I/O map, as well as the property of fading memory, which roughly translates to the property that the reservoir forgets initial conditions and thus depends more strongly on more recent inputs (For instance, for multi-stable dynamical systems, a global representation such as Eq. (1) may not exist. However, a local representation around each steady state can be shown to exist with a finite convergence radius). The realization of such a time-invariant map is essential for a physical system to be reliably employed for inference on an input signal of arbitrary length, and thus for online time series processing.

In classical physical systems, the existence of a unique information steady state and the resulting fading memory property is determined only by the input encoding dynamics – the map from input series to system state. More explicitly, the information extraction step (sometimes referred to as the "output layer") on a classical system is considered to be a passive action, so that the state can always be observed at the precision required. However, for physical systems operating in the quantum regime, the role of quantum measurement is fundamental: in addition to the inherent uncertainty in quantum measurements as dictated by the Heisenberg uncertainty principle, the conditional dependence of the statistical system state on prior measurement outcomes – referred to as backaction – strongly determines the information that can be extracted. Recent work in circuit-based quantum computation has shown that the qualitative features of the statistical steady state of monitored circuits strongly depend on the rate of measurement[46,47]. In particular, generic quantum systems that alternate dynamics and measurement (input encoding and output in the present context) are known to give rise to deep thermalization of the memory subsystem[48,49], resulting in an approximate Haar-random state with vanishing temporal memory. The absence of a comprehensive framework in QML for analyzing and implementing an encoding-decoding system with finite temporal memory, along with characterization tools for the accessible set of input-output functionals, has hindered both a systematic study and the practical application of online learning methods.

Here, we develop both a general temporal learning framework suitable for qubit-based quantum processors and the associated methods of analysis based on an appropriate generalization of the Volterra Series analysis to monitored quantum systems, the Quantum Volterra Theory (QVT). Our approach incorporates the effects of backaction that results from quantum measurements in the process of information extraction.

We begin by providing a fundamental description of both the information input and output processes that enable general time series processing with quantum systems before specializing in the NISQRC algorithm. The 'input' component of the map is given by a pipeline (encoding) that injects temporal data $\{u_n\}$ into a quantum system through a general parameterized quantum channel $\mathcal{U}(u_n)\hat{\rho}$. This channel could describe for instance continuous Lindblad evolution for a duration $\tau$, namely $e^{\tau \mathcal{L}(u_n)}\hat{\rho}$, as in Results' subsection "Practical machine learning using temporal data", or a discrete set of gates as in Results' subsection "Experimental results on the quantum system"; $\mathcal{U}(u_n)$ is generally applied to all qubits, and we assume only that they are not explicitly monitored for its duration.

To enable persistent memory in the presence of quantum measurement, we separate the $L$-qubit system into $M$ memory qubits and $R$ readout qubits ($L = M + R$) and denote their respective Hilbert spaces with superscript M and R. After evolution under any input $u_n$, only the $R$ readout qubits are (simultaneously) measured; this separation therefore allows for the concept of partial measurements of the full quantum system, which proves critical to the success of NISQRC. The measurement scheme itself can be very general, characterized by a positive operator-valued measure (POVM)

$$\mathcal{O}_R = \left\{ \hat{M}_j | \hat{M}_j = \hat{I}^{\otimes M} \otimes \hat{E}_j \right\} \tag{2}$$

satisfying $\hat{E}_j \succcurlyeq 0$ and $\sum_j \hat{E}_j = \hat{I}^{\otimes R}$. Here we will consider a practically implementable measurement in the readout qubit computational basis, described by $\hat{E}_j = |\mathbf{b}_j\rangle \langle \mathbf{b}_j|$: each bit-string $\mathbf{b}_j$ is the $R$-bit binary representation of integer $j \in \{0, 1, \cdots, 2^R - 1\}$ denoting the bit-wise state of the measured qubits.

As elucidated by the QVT analysis of Results' subsection "Quantum Volterra Theory", a purification mechanism must necessarily accompany readout to prevent thermalization and furnish our quantum architecture with persistent fading memory. This is accomplished by following each projective measurement operation with a deterministic reset to the ground state $|0\rangle$. The resulting measure-reset operation we employ throughout this paper is formally described by the POVM operators $\hat{E}_j = \hat{K}_j^\dagger \hat{K}_j$ in Eq. (2), with non-hermitian Kraus operators $\hat{K}_j = |\mathbf{b}_0\rangle \langle \mathbf{b}_j|$. In each measure-reset step, only the readout qubits are measured in the computational basis and then reset to the ground state, irrespective of the measurement outcome.

NISQRC is distinguished by the iterative encode-measure-reset scheme depicted in Fig. 1. Explicitly, for a given input sequence $\mathbf{u}$ with length $N$, we initialize the system in the state $\hat{\rho}_0^M \otimes |0\rangle \langle 0|^{\otimes R}$. For each element of the input sequence $u_n$, an encoding step is comprised of unmonitored evolution of all qubits via $\mathcal{U}(u_n)$ followed by a measure-reset operation $\mathcal{O}_R$. The measurement outcome in this single shot is a random bitstring $\mathbf{b}^{(s)}(n)$, and the resulting state is $\hat{\rho}_n^{M,\text{cond}} \otimes |0\rangle \langle 0|^{\otimes R}$: the memory qubits are in a state conditioned on the measurement outcome, and the readout qubits are reset. The subsequent input is then encoded in this state, i.e., $\mathcal{U}(u_{n+1}) \left( \hat{\rho}_n^{M,\text{cond}} \otimes |0\rangle \langle 0|^{\otimes R} \right)$, and the process is iterated as long as there is data in the pipeline. This structure elucidates the naming of the unmeasured memory qubits: these are the only qubits that retain the memory of past inputs.

The above description yields a set of $N$ measurement outcomes $\{\mathbf{b}^{(s)}(n)\}$ observed in a single shot $s$ of the quantum circuit. In order to obtain statistics and therefore to output features as expected values of observables $\hat{M}_j$, we perform $S$ repetitions of this circuit for a given $\mathbf{u}$

sequence: the total execution time is $NS$, linear with respect to shots $S$ and input length $N$. The resulting readout features are formally defined as the probability

$$x_j(n) = \Pr[\mathbf{b}^{(s)}(n) = \mathbf{b}_j | \mathbf{u}], \qquad (3)$$

which are estimated by the empirical mean $\bar{x}_j(n) = (1/S)\sum_s \delta(\mathbf{b}^{(s)}(n), \mathbf{b}_j)$ of $\{\mathbf{b}^{(s)}(n)\}_{s \in [S]}$ (see "Methods" subsection "Generating features via conditional evolution and measurement" for more details of the NISQRC algorithm). We show in Supplementary Note 2 that at time step $n$, $x_j(n)$ can be computed efficiently through

$$x_j(n) = \mathrm{Tr}\left(\hat{M}_j \hat{\rho}_n^{\mathrm{MR}}\right), \qquad (4)$$

where $\hat{\rho}_n^{\mathrm{MR}}$ is the effective full $L$-qubit system state at time step $n$ prior to measurement.

The output $y_n \equiv \mathbf{y}(n) = \mathbf{w} \cdot \mathbf{x}(n) \in \mathbb{R}$ is obtained from the measurement results in each step, defining the functional I/O map which we characterize next (see details in "Methods" subsection "Generating features via conditional evolution and measurement" and "The quantum Volterra theory and analysis of NISQRC"). This complete architecture, from the quantum circuit generating measurement outcomes for a given input, to the construction of weighted output features, is depicted schematically in Fig. 1. We note that reset operations have been used implicitly in prior work on quantum reservoir algorithms, where the successive inputs are encoded in the state of an 'input' qubit[26,50]. However the critical role of the reset operation in endowing a quantum reservoir with a persistent memory, discussed in the next subsection, has so far not been highlighted.

While for null inputs (i.e., $u_n = 0$ for all $n$) such quantum systems are guaranteed to have a unique statistical steady state, the existence of a nontrivial memory and kernel structure is much more involved. Through QVT (see "Methods" subsection "The quantum Volterra theory and analysis of NISQRC"), we show that these requirements place strong constraints on the encoding and measurement steps viz. the choice of $(\mathcal{U}, \hat{M}_j)$. This then enables us to propose an algorithm for online learning that provably provides a controllable and time-invariant temporal memory (which will be referred to as *persistent memory*) – enabling inference on arbitrarily long input sequences even on NISQ hardware without any error mitigation or correction.

## Quantum Volterra theory

In NISQRC the purpose of the partial reset operation is to endow the system with asymptotic time-invariance, a finite persistent memory, and a nontrivial Volterra Series expansion for the system state (see "Methods" subsection "The quantum Volterra theory and analysis of NISQRC" and Supplementary Note 3):

$$\hat{\rho}_n^{\mathrm{MR}} = \sum_{k=0}^{\infty} \sum_{n_1=0}^{\infty} \cdots \sum_{n_k=n_{k-1}}^{\infty} \hat{h}_k(n_1, \cdots, n_k) \prod_{\kappa=1}^{k} u_{n-n_\kappa}. \qquad (5)$$

where all Volterra kernels $\hat{h}_k$ are quantum operators. The classical kernels in Eq. (1) describing the *measured features* can be extracted through $h_k^{(j)} = \mathrm{Tr}(\hat{M}_j \hat{h}_k)$. We refer to this analysis as the Quantum Volterra Theory (QVT). Through analytical arguments based on the QVT, we show that omitting the partial reset operation renders all Volterra kernels trivial – a finding corroborated by our experimental results in Results' subsection "Experimental results on quantum system".

QVT also provides a way to characterize the important memory time-scales of the I/O map generated by the NISQRC algorithm through a given encoding, which we use in Results' subsection "Practical machine learning using temporal data" to aid encoding design for a specific ML task on an experimental system. In what follows, we show

that inference on an indefinitely long input sequence can be done even in the presence of dissipation and decoherence.

Consider an input-encoding $\mathcal{U}(u_n)\hat{\rho} = e^{\tau \mathcal{L}(u_n)}\hat{\rho}$ where

$$\mathcal{L}(u)\hat{\rho} = -i[\hat{H}(u), \hat{\rho}] + \mathcal{D}_{\mathrm{T}}\hat{\rho}, \qquad (6)$$

representing evolution under a parameterized Hamiltonian $\hat{H}(u_n)$ for a duration $\tau$ in the presence of dissipation $\mathcal{D}_{\mathrm{T}}$. For concreteness, we take $\mathcal{D}_{\mathrm{T}} = \sum_{i=1}^{L} \gamma_i \mathcal{D}[\hat{\sigma}_i^{-,z}]$ describing decoherence processes and study here a specific Ising Hamiltonian encoding $\hat{H}(u) = \hat{H}_0 + u \cdot \hat{H}_1$ inspired by quantum annealing and simulation architectures (other ansätze can likewise be considered),

$$\hat{H}_0 = \sum_{\langle i,i' \rangle} J_{i,i'} \hat{\sigma}_i^z \hat{\sigma}_{i'}^z + \sum_{i=1}^{L} \eta_i^x \hat{\sigma}_i^x, \hat{H}_1 = \sum_{i=1}^{L} \eta_i^z \hat{\sigma}_i^z. \qquad (7)$$

The coupling strength $J_{i,i'}$, transverse $x$-field strength $\eta_i^x$ and longitudinal $z$-drive strength $\eta_i^z$ are randomly chosen but then fixed for all inputs $\{u_n\}$ (see Supplementary Note 1 for more details). The encoding channel is applied for duration $\tau$, and each qubit has a finite lifetime $T_1 = \gamma^{-1}$. We will specify the number of memory and readout qubits of a given QRC with the notation $(M + R)$.

In Fig. 2(a), we plot the first two Volterra kernels $h_1$ and $h_2$ (cf. Eq. (1)), for a random $(2+1)$-qubit QRC using the above encoding and the reset scheme. The expression for these kernels has been derived from the QVT; their numerical construction is discussed in Methods, also see Supplementary Equations 43–45. Importantly, we find all kernels have an essential dependence on the statistical steady state or fixed-point in the absence of any input: $\hat{\rho}_{\mathrm{FP}}^{\mathrm{M}} = \lim_{n \to \infty} \hat{\rho}_n^{\mathrm{M}}|_{u_n = 0}$. Here $\hat{\rho}_n^{\mathrm{M}}|_{u_n = 0} = \mathcal{P}_0^n \hat{\rho}_0^{\mathrm{M}}$ is obtained by $n$ applications of the null-input single-step quantum channel $\mathcal{P}_0$, defined in Methods' subsection "The quantum Volterra theory and analysis of NISQRC". The properties of quantum Volterra kernels, including their characteristic decay time, can be related to the spectrum of $\mathcal{P}_0$, defined by $\mathcal{P}_0 \hat{\varrho}_\alpha^{\mathrm{M}} = \lambda_\alpha \hat{\varrho}_\alpha^{\mathrm{M}}$. Here $\hat{\varrho}_\alpha^{\mathrm{M}}$

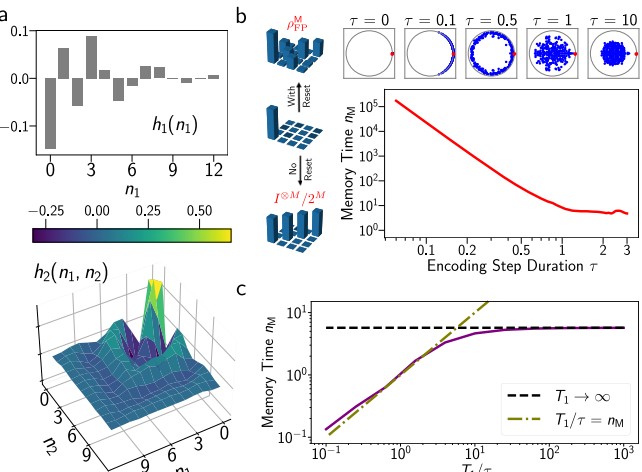

**Fig. 2 | Quantum Volterra Theory (QVT) analysis for $(M + R)$-qubit reservoir. a** First and second order Volterra kernels in a $(2+1)$-qubit QRC, which vanish at large $n_1$ and $n_2$ due to finite memory $n_{\mathrm{M}}$. **b** Fixed-point of memory subsystem $\hat{\rho}_{\mathrm{FP}}^{\mathrm{M}}$ with reset (top) and without reset (bottom), starting from an arbitrary initial state (center). Without reset, the fixed point is always the trivial fully-mixed state and Volterra kernels vanish. The top panel shows the distribution of the $4^M = 256$ eigenvalues of $\mathcal{P}_0$ in a $(4+2)$-qubit QRC, where red dots correspond to the static unit eigenvalue $\lambda_1 = 1$. The remaining eigenvalues $\lambda_{\alpha \geq 2}$ (blue) evolve with evolution time $\tau$, leading to a variable memory time. The bottom panel shows the resulting memory time $n_{\mathrm{M}}$ as a function of the evolution duration $\tau$. **c** Memory time $n_{\mathrm{M}}$ as a function of qubit lifetimes $T_1 = \gamma^{-1}$, in terms of the evolution duration $\tau$ in a $(4+2)$-qubit QRC. Provided $T_1 \gg \tau$, $n_{\mathrm{M}} \to n_{\mathrm{M}}^0$, so that the QRC memory is mostly dominated by its lossless dynamical map and not by $T_1$ in this regime.

are eigenvectors that exist in the $4^M$-dimensional space of memory subsystem states. The eigenvalues satisfy $1 = \lambda_1 \geq |\lambda_2| \geq \cdots \geq |\lambda_{4^M}| \geq 0$; examples are plotted in Fig. 2(b) for various values of $\tau$. The unique eigenvector corresponding to the largest eigenvalue $\lambda_1 = 1$ is special, being the fixed point of the memory subsystem, $\hat{\varrho}_1^M = \hat{\rho}_{FP}^M$, reached once transients have died out.

The second largest eigenvalue $\lambda_2$ determines the time over which memory of an initial state persists as this fixed point is approached, and is used to identify a *memory time* $n_M = -1/\ln|\lambda_2|$. Note that this quantity is dimensionless and can be converted to the actual passage of time through multiplication by $\tau$, while $n_M$ itself non-trivially depends on $\tau$ (see Fig. 2(b)). Memory time describes an effective 'envelope' for a system's Volterra kernels; an additional nontrivial structure is also required for QRC to produce meaningful functionals of past inputs. With the spectral problem at hand, we next analyze the information-theoretical benefit of the reset operation. Firstly, the absence of the unconditional reset operation produces a *unital* $\mathcal{P}_0$ ("unital" refers to an operator that maps the identity matrix to itself) with resulting $\hat{\rho}_{FP}^M = I^{\otimes M}/2^M$. This fully mixed state is inexorably approached after $n_M$ steps under any input sequence and retains no information on past inputs: all Volterra kernels, therefore, vanish, despite a generally-finite $n_M$. Such algorithms (e.g., ref. 30) are only capable of processing input sequences of length $n_M$ and would not retain a persistent memory necessary for inference on longer sequences of inputs. Hence such encodings would be unsuitable for online learning on streaming data. The possibility of inference through the transients has been observed and utilized before (see e.g., refs. 12,51,52) in the context of classical reservoir computing. However, the simple yet essential inclusion of the purifying reset operation avoids unitality – more generally, a common fixed point for all *u*-encoding channels – which we find is the key to enabling nontrivial Volterra kernels and consequent online QRC processing (see "Methods" subsection "The quantum Volterra theory and analysis of NISQRC" and also ref. 53). Once such an I/O map is realized, $\lambda_\alpha$ and the consequent memory properties can be meaningfully controlled by the QRC encoding parameters. As shown in Fig. 2(b) the characteristic decay time set by $n_M$, for instance, decreases across several orders-of-magnitude with increasing $\tau$.

The partial measurement and reset protocol also resolves the unfavorable quadratic runtime scaling of prior approaches. A wide range of proposals and implementations of QRC[27,29,54] consider the read-out of all constituent qubits at every output step, terminating the computation. Not only does this preclude inference on streaming data, it requires the entire input sequence to be re-encoded to proceed one step further in the computation, leading to an $O(N^2 S)$ running time. As shown in schematic Fig. 1, incorporating partial measurement with reset in NISQRC does not require such a re-encoding; the entire input sequence can be processed in any given measurement shot $S$, enabling online processing with an $O(NS)$ runtime, while maintaining a controllable memory timescale. We note that an alternative scheme to remedy this issue has been suggested in ref. 50, which relies on information extraction through continuous weak measurement.

Next, we show that the nontrivial nature of Volterra kernels realized by the NISQRC algorithm is preserved under the inclusion of dissipation. For example, we explore the effect of finite qubit $T_1$ on $n_M$ in Fig. 2(c). If $T_1/\tau > n_M^0$, where $n_M^0$ is the memory time of the lossless map, then $n_M \to n_M^0$ and is essentially independent of $T_1$, determined instead by the unitary and measurement-induced dynamics. Therefore the design of the encoding algorithm has to be guided by matching the memory time of the reservoir to the longest correlation time in the input data. Additional design criteria are discussed in Section Discussion. As a result, lossy QRCs can still be deployed for online processing, with a total run time $T_{run}$ that is unconstrained by (and can therefore far exceed) $T_1$. We will demonstrate this via simulations in Results' subsection "Practical machine learning using temporal data" with

$T_{run} \gg T_1$, and via experiments in Results' subsection "Experimental results on the quantum system" for $T_{run} \simeq T_1$; in the latter $T_{run}$ is limited only by memory buffer constraints on the classical backend.

## Practical machine learning using temporal data

Thus far, we have assumed outputs to be expected features $x_j(n)$, which, in principle, assumes an infinite number of measurements. In any practical implementation, one must instead estimate these features with $S$ shots or repetitions of the algorithm for a given input **u**. The resulting QSN constrains the learning performance achievable in experiments on quantum processors in a way that can be fully characterized[9] and is therefore also included in numerical simulations which we present next.

To demonstrate the utility of the NISQRC framework, we consider a practical application of machine learning on time-dependent classical data: the *channel equalization* (CE) task[11,14]. Suppose one wishes to transmit a message $m(n)$ of length $N$, which here takes discrete values from $\{-3, -1, 1, 3\}$, through an unknown noisy channel to a receiver. This medium generally distorts the signal, so the received version $u(n)$ is different from the intended $m(n)$. Channel equalization seeks to reconstruct the original message $m(n)$ from the corrupted signal $u(n)$ as accurately as possible, and is of fundamental importance in communication systems. Specifically, we assume the message is corrupted by nonlinear receiver saturation, inter-symbol interference (a linear kernel), and additive white noise[11,14] (additional details in Supplementary Note 6). As shown in Fig. 3(a), even if one has access to the exact inverse of the resulting nonlinear filter, the signal-to-noise (SNR) of the additive noise bounds the minimum achievable error rate. We also show the error rates of simply rounding $u(n)$ to the nearest $m$, and a

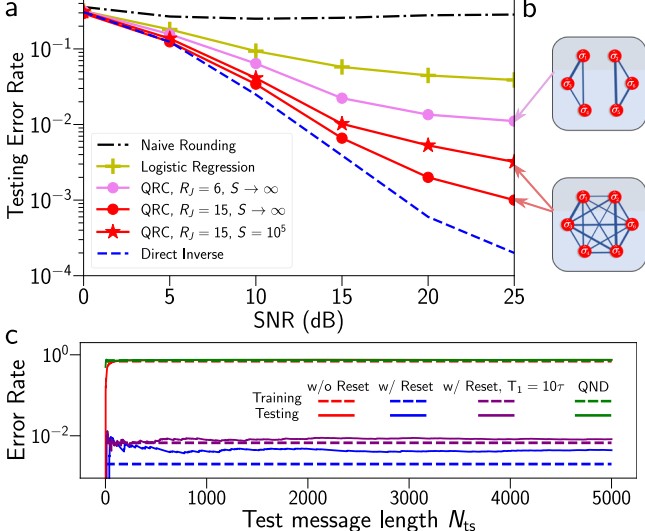

**Fig. 3 | Numerical results for the channel equalization (CE) task with Hamiltonian ansatz. a** Error rates on test messages for the CE task with a Hamiltonian ansatz (2 + 4)-qubit QRC for two distinct connectivities shown in (**b**) The fully-connected QRC in red has Jacobian rank $R_J = 2^R - 1 = 15$ and is shown for both $S \to \infty$ (circles) and finite $S = 10^5$ (★), whereas the split QRC has $R_J = 2(2^2 - 1) = 6$ and only $S \to \infty$ is plotted in magenta. These are compared with the error rates of naive rounding (black dash-dots) and logistic regression on the current signal (yellow +, see Supplementary Note 6), and the exact channel inverse (blue dashed). **c** Performance of connected QRC on SNR = 20dB test signals (solid) of increasing length $N_{ts} \leq 5000$, with shots $S = 10^5$. Training error on $N = 100$-length messages is indicated for comparison in dashed lines. Without reset (red) or using 4 ancilla qubit ansatz with quantum non-demolition (QND) readout (proposed in ref. 30, green), the algorithms both fail, approaching the random guessing error rate and showing that both architectures suffer from the thermalization problem. Performance is only slightly reduced from the dissipation-free case (blue) when strong decay $T_1 = 10\tau$ is included (purple). All error rates in (**c**) are averaged over 8 different test messages.

direct logistic regression on $u(n)$ (i.e., a single-layer perceptron with a softmax activation – see Supplementary Note 6). for comparison. Both these approaches are linear and memory-less and, therefore, perform poorly on the non-trivial nonlinear filter we consider, although logistic regression outperforms rounding ($\approx 30\%$) by inverting the linear portion of the distortion.

We now perform the CE task using the NISQRC algorithm on a simulated (2 + 4)-qubit reservoir under the ansatz of Eq. (7), as could be realized in quantum annealing hardware (see Fig. 3). We will later demonstrate the same task in experiments with a completely different quantum system and encoding ansatz, implemented on a superconducting quantum processor (see Fig. 4). The ability to efficiently compute the Volterra kernels for this quantum system immediately provides guidance regarding parameter choices. In particular, we choose random parameter distributions such that the average (across the circuit) $J_{i,i'}\tau$, $\eta_i^x\tau$ and $\eta_i^z\tau$ provides a memory time $n_M \approx O(10^1)$, on the order of the length of the distorting linear kernel $h(n)$, which is 8. These QRCs have $K = 2^4 = 16$ readout features $\{x_j(n)\}_{j\in[K]}$ whose corresponding time-independent output weights **w** are learned by minimizing cross-entropy loss on 100 training messages of length $N = 100$ (see Supplementary Note 6 for additional details). The resulting NISQRC performance on test messages is studied in Fig. 3(a), where we compare two distinct coupling maps shown in Fig. 3(b). In the highly-connected (lower) system the performance approaches the theoretical bound for $S \to \infty$; finite sampling (here, $S = 10^5$ is in the range typically used in experiments) increases the error rate as expected, but the

increase in error rate in numerical simulations is observed to depend on the encoding (not reported here). In all cases, NISQRC significantly outperforms direct logistic regression due to its ability to reliably implement non-linear memory kernels and therefore approximate the distorting channel inverse.

We note that the split system (upper) performs significantly worse even without sampling noise: this is because the quantum system lives in a smaller effective Hilbert space – the product of two disconnected three-qubit systems – and is far less expressive as a result. Although in both cases the number of measured features is the same, those from the connected system span a richer and independent space of functionals. This functional independence can be quantified by the Jacobian rank $R_J$, which is the number of independent **u**-gradients that can be represented by a given encoding (Supplementary Note 5); an increased connectivity and complexity of state-description generally manifests as an increase in the Jacobian rank and consequent improved CE task performance. This observation can be viewed as a generalization of the findings in time-independent computation[9] to tasks over temporally-varying data, and also agrees with related recent theoretical work[29].

Most importantly, we demonstrate in Fig. 3(c) that the NISQRC algorithm enables the use of a quantum reservoir for online learning. In all cases studied here, $N = 100$ is used for training and the length of the SNR = 20dB test messages $N_{ts}$ is varied. As suggested by the QVT, the performance is unaffected by $N_{ts}$ even if it greatly exceeds the lifetime of individual qubits: $N_{ts} = T_{run}/\tau \gg T_1/\tau = 10$, and NISQRC can, therefore, be used to perform inference on an indefinite-length signal with noisy quantum hardware. As seen in the same figure, while dissipation imposes only a small constant performance penalty, the reset operation is critical: if removed, the error rate returns to that of random guessing, as the Volterra kernels vanish and the I/O map becomes trivial.

We finally note that an arbitrarily-inserted reset operation may not be sufficient to create a non-zero persistent memory. For instance, an analysis based on the QVT shows that despite its use in a recently studied reservoir algorithm[30] (based on a quantum non-demolition measurement proposal in ref. 27), the reset operation can not avoid a zero persistent memory, effectively resulting in an amnesiac reservoir. In this scheme, the quantum circuit is coupled to ancilla qubits by using transversal CNOT gates. Upon closer examination it is found that while the projective measurement of ancilla qubits leads to read out of system qubits and their collapse to the ancilla state via backaction, subsequent reset of the ancillas does not reset the system qubits. This scheme therefore suffers from the same thermalization problem as any no-reset NISQRC does, and hence has zero persistent memory. We verify this analysis in Fig. 3(c) by implementing the CE task with a four-ancilla-qubit circuit. The error rates are found to be very close to the no-reset-NISQRC one, whose I/O map we have shown before to be trivial (see also Fig. 3(c)).

### Experimental results on quantum systems

We now demonstrate NISQRC in action by performing the SNR = 20dB CE task on an IBM Quantum superconducting processor. To highlight the generality of our NISQRC approach, we now consider a circuit-based parametric encoding scheme inspired by a Trotterization of Eq. (7), suitable for gate-based quantum computers. In particular, we use a $L = 7$ qubit linear subgraph of the *ibm_algiers* device, with $M = 3$ memory qubits and $R = 4$ readout qubits in alternating positions, as depicted in Fig. 4(a). The encoding unitary for each time step $n$ is also shown:

$$\hat{U}(u_n) = \left( \mathcal{W}(J)\mathcal{R}_z(\boldsymbol{\theta}^z + \boldsymbol{\theta}^I u_n)\mathcal{R}_x(\boldsymbol{\theta}^x) \right)^{n_T},$$

where $\mathcal{R}_{x,z}$ are composite Pauli-rotations applied qubit-wise, and $\mathcal{W}(J)$ defines composite $\mathcal{R}_{zz}$ gates between neighboring qubits, all repeated $n_T = 3$ times (for parameters $\boldsymbol{\theta}^{x,z,I}$, $J$ and further details see "Methods" subsection "IBMQ implementation").

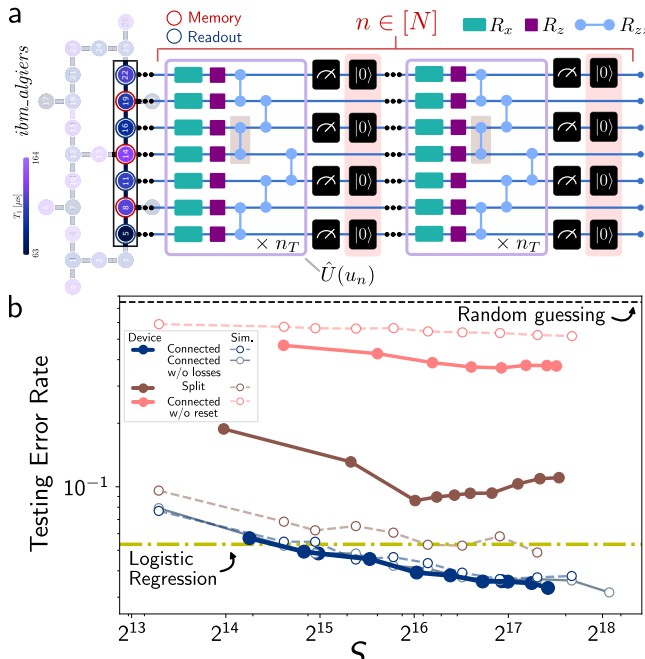

**Fig. 4 | Experimental results for the channel equalization task with circuit ansatz. a** (3 + 4)-qubit linear chain of the *ibm_algiers* device used to perform the CE task. Filled colors represent qubit $T_1$ time according to the displayed colorbar, for the specific experimental run with the split chain. Qubits indexed {8, 14, 19} are used for memory and qubits {5, 11, 16, 22} for readout, and gate-decomposition of the encoding unitary $\hat{U}(u_n)$ is depicted. Removing gates shaded in brown yields two smaller chains to explore the role of connectivity, while removing reset operations (shaded peach) allows switching from a non-unital to a unital I/O map. **b** Testing error rates for the SNR = 20dB CE task of Results' subsection "Practical machine learning using temporal data" with $N = 20$ on the *ibm_algiers* device in filled circles and in simulation in open circles, as a function of number of shots $S$. The connected circuit in blue outperforms the split circuit in brown and the circuit without reset in peach. For comparison, we plot the testing error rate of logistic regression (yellow line), as well as random guessing (black dashed line).

Realizing the NISQRC framework with the circuit ansatz depicted in Fig. 4(a) requires the state-of-the-art implementation of mid-circuit measurements and qubit reset, which has recently become possible on IBM Quantum hardware[55]. We plot the testing error using the indicated linear chain of the *ibm_algiers* device as a function of the number of shots $S$ in solid blue Fig. 4(b), alongside simulations of both the ideal unitary circuit and with qubit losses in open circles. We clearly observe that performance is influenced by the number of shots available, and hence by QSN. In particular, for a sufficiently large $S$, the device outperforms the same logistic regression method considered previously. For the circuit runs, the average qubit coherence times over 7 qubits are $T_1^{av} = 124\,\mu s$, $T_2^{av} = 91\mu s$ (see Supplementary Note 9 for the ranges of all parameters, which vary over the time of runs as well), while the total circuit run time for a single message is $T_{run} \approx 117\,\mu s$. Even though $T_{run} \simeq T_1^{av}$, the CE task performance using NISQRC on *ibm_algiers* is essentially independent of qubit lifetimes. This is emphatically demonstrated by the excellent agreement between the experimental results and simulations assuming infinite coherence-time qubits. In fact, finite qubit decay consistent with *ibm_algiers* leaves simulation results practically unchanged (as plotted in dashed blue); we find that $T_1$ times would have to be over an order of magnitude shorter to begin to detrimentally impact NISQRC performance on this device (see Supplementary Note 7). We further find that artificially increasing $T_{run}$ beyond $T_1$ by introducing controlled delays in each layer also leaves performance unchanged (see Supplementary Note 8).

Using the same device, we are able to analyze several important aspects of the NISQRC algorithm. First, we consider the same CE task with a split chain, where the connection between the qubits labeled '14' and '16' on *ibm_algiers* is severed by removing the $R_{zz}$ gate highlighted in brown in Fig. 4(a). The resulting device performance using these two smaller chains is worse, consistent both with simulations of the same circuit and the analogous split Hamiltonian ansatz studied in Results' subsection "Practical machine learning using temporal data". Next, we return to the 7 qubit chain but now remove reset operations in the NISQRC architecture, shaded in red in Fig. 4(a): all other gates and readout operations are unchanged. The device performance now approaches that of random guessing: the absence of the crucial reset operation leads to an amnesiac QRC with no dependence on past or present inputs. This remarkable finding reinforces that reset operations demanded by the NISQRC algorithm are, therefore, essential to imbue the QRC with memory and enable any non-trivial temporal data processing.

We note that for these experiments, while performance qualitatively agrees well with simulations, some quantitative discrepancies are observed. Our deployment of mid-circuit measurements in their earliest implementation on IBM Quantum was accompanied by some technical constraints; for example, not all shots for a given instance of the CE task could be collected in contiguous repeated device runs, instead sometimes being separated by several hours (due to queuing times as well as classical memory buffer constraints on the number of shots that could be collected in a single experiment). Simply put, this means that the device could suffer non-trivial parameter drifts from one type of device configuration to the next, and even during the course of collecting all shots for a specific configuration. In particular, we find that qubit lifetimes for experiments with the split chain, and the connected chain without reset, were significantly shorter than for the connected chain with reset (see Supplementary Tables 1–3), which could lead to the discrepancy in comparison to simulations, where we assumed a fixed coherence time distribution. Resource constraints similarly restrict us to limited training and testing set sizes, which can also lead to variance in performance. We anticipate such technical constraints to be alleviated as mid-circuit measurement implementations mature on IBM Quantum, enabling even more accurate correspondence with simulations.

We also note that there is room for improvement in CE performance when compared against Hamiltonian ansatz NISQRC of similar scale in Fig. 3. A key difference is the reduced number of connections in the nearest-neighbor linear chain employed on *ibm_algiers*; including effective $R_{zz}$ gates between non-adjacent qubits significantly increases the gate-depth of the encoding step, enhancing sensitivity to circuit-fidelity due to increasing runtimes. The circuit ansatz can also be optimized - using knowledge of the Volterra kernels - for better nonlinear processing capabilities demanded by the CE task, in addition to memory capacity determined by $n_M$. Nonetheless, the demonstrated performance and robustness of the NISQRC framework to dissipation already suggests its viability for increasingly complex time-dependent learning tasks using actual quantum hardware.

## Discussion

A key technical advancement in our work is the formulation of the Quantum Volterra Theory (QVT) to describe the time-invariant input-output map of a quantum system under temporal inputs and repeated measurements. Insights provided by the QVT enabled us to propose the essential component of the NISQRC algorithm - deterministic post-measurement reset to avoid thermalization due to repeated measurements - which allows the quantum system to retain persistent memory of temporal inputs even under projective measurements and their associated strong backaction. The resulting algorithm enables inference on a signal that can be arbitrarily long, provided the encoding is designed to endow the reservoir with a memory that matches the longest correlation time in the data.

While we have applied the QVT to qubit-based circuits, our analysis does not make an explicit assumption on the Hilbert space dimension of the quantum system, and allows for completely general measurements through its formulation in terms of POVMs; as a result, it can be applied to other finite-level quantum systems such as qudits[24], and can be extended to continuous-variable quantum systems[28,31]. We, therefore, believe the QVT provides the ideal framework to analyze the memory and computational capacity of temporal information processing schemes using general quantum systems and their associated measurement protocols. We note here that the use of continuous weak measurements, analyzed in ref. 50, provides an alternative approach to endowing the reservoir with finite persistent memory and can be analyzed with QVT for its task-specific optimization.

Going beyond the crucial reset component, we have demonstrated that QVT can be invaluable in identifying general design principles for qubit-based systems as reservoirs. For example, while measuring some fraction of qubits is essential for extracting information, measuring all qubits imposes a trivial memory time. We employ $M \simeq R$ in this work, but an optimal separation of memory and readout qubits may depend on specific tasks. A simple rule of thumb is to choose $M/R$, together with $\tau$ and other drive strengths, to match the memory time of the physical system to the longest correlation time in the data. In the channel equalization task studied in Results' subsection "Practical machine learning using temporal data", the data correlation time is fixed by choice of the distorting channel and we have then chosen $M/R$ to endow the quantum system with a memory time – calculated through the QVT formalism – that matches that time, about 8 steps (recall memory time is measured in the number of encoding steps). Especially, the duration of the unmonitored dynamics, $\tau$, has been chosen to be long enough to generate non-linear kernels that match the known order of the non-linearity of the distorting channel, but short enough to avoid limiting the memory time by the shortest $T_1$. For the latter requirement, the kind of analysis shown in Fig. 2c, calculated through QVT, can act as a very helpful guide. We have observed that even when there is a large spread in $T_1$, the physical memory time may be longer than the shortest $T_1$, presumably through the delocalization of the information on longer-lived memory qubits in the circuit. In addition, qubit connectivity, analyzed in Fig. 3a, can help

with the generation of functions that are sufficiently complex to match the functional complexity of the task.

Finally, the most crucial design criterion for any quantum system intended to process streaming data is that the map $\mathcal{P}_0$ be non-unital. In an architecture with memory and readout qubits, the presence of a reset operation is essential but not on its own sufficient: as noted earlier the quantum channel must additionally contain input-dependent operations on both memory and readout qubits to prevent scrambling of memory qubits and endow the QRC state with the fading memory property. To address an important example, it is straightforward to confirm that any channel with input-dependent operations on only memory qubits $\mathcal{U}(u_n)$ and an arbitrary set of controlled gates from memory to readout qubits (e.g., Fig. 5d in ref. 30) is unital on memory qubits and therefore lacks a persistent temporal memory. We note that in such cases the reservoir can still be trained to implement complicated functions in the transient state[12], but genuine online learning will not be possible. The QVT presented here prescribes how to avoid such pitfalls when designing a quantum channel for temporal data processing: one can simply check whether the resulting $\mathcal{P}_0$ is a unital map. We have not carried out here an exhaustive study of the optimal design principles for more complex or general classes of tasks, but we hope that the simple and fundamental guidelines we have followed for designing an experimental reservoir to accurately carry out equalization on RF-encoded messages illustrate the utility of QVT in the design of a hardware reservoir.

By enabling online learning in the presence of losses, NISQRC paves the way to harness quantum machines for temporal data processing in far more complex applications than the CE task demonstrated here. Examples include spatiotemporal integrators, and ML tasks where spatial information is temporally encoded, such as video processing. Recent results provide evidence that the most compelling applications, however, lie in the domain of machine learning on weak signals originating from other, potentially complex quantum systems[15,56], for the purposes of quantum state classification. In tackling such increasingly complex tasks, the scale of quantum devices required is likely to be larger than those employed here. The NISQRC framework can be applied irrespective of device size; however, its readout features at a given time live in a $K = 2^R$ dimensional space. For applications requiring a large $R$, the exponential growth of the feature-space dimension may give rise to concerns with under-sampling, as in practice the available number of shots $S$ may not be sufficiently large. In such large-$R$ regimes, certain linear combinations of measured features can be found, known as *eigentasks*, that provably maximize the SNR[9] of the functions approximated by a given physical quantum system trained with $S$ shots. Eigentask analysis provides very effective strategies for noise mitigation. In ref. 9 the *Eigentask Learning* methodology was proposed to enhance generalization in supervised learning. For the present work, such noise mitigation strategies were not needed as the size of the devices used was sufficiently small to efficiently sample. An interesting direction is the application of Eigentask analysis to NISQRC, which we leave to future work.

The present work and the availability of an algorithm for information processing beyond the coherence time presents new opportunities for mid-circuit measurement and control. While mid-circuit measurement is essential for quantum error correction[57], its recent availability on cloud-based quantum computers has allowed the exploration of other quantum applications on near-term noisy qubits. Local operations such as measurement followed by classical control for gate teleportation have been used to generate nonlocal entanglement[58–60]. In addition, mid-circuit measurements have been employed to study critical phenomena such as phase transitions[61–63] and are predicted to allow nonlinear subroutines in quantum algorithms[64]. The present work opens up a new direction in this application space, namely the design of self-adapting circuits for inference on temporal

data with slowly changing statistics. This would require dynamic programming capabilities for mid-circuit measurements, not employed in the present work. We show here that implementing even the relatively simple CE task challenges current capabilities for repeated measurements and control; having the means to deploy more complex quantum processors for temporal learning via NISQRC can push hardware advancements to more tightly integrate quantum and classical processing for efficient machine-based inference.

## Methods

### Generating features via conditional evolution and measurement

Here we detail how an input-output functional map is obtained in the NISQRC framework. The quantum system is initialized to $\hat{\rho}_0^{\mathrm{MR}} = \hat{\rho}_0^{\mathrm{M}} \otimes |0\rangle \langle 0|^{\otimes R}$, where $\hat{\rho}_0^{\mathrm{M}}$ is the initial state, which is usually set to be $|0\rangle \langle 0|^{\otimes M}$. Then, for each run or 'shot' indexed by $s$, the process described in the following paragraph is repeated.

Before executing the $n$-th step, the overall state can be described as $\hat{\rho}_{n-1}^{\mathrm{M,cond}} \otimes |0\rangle \langle 0|^{\otimes R}$ (usually pure), where the superscript cond emphasizes that the memory subsystem state is generally conditioned on the history of all previous inputs $\{u_m\}_{m \le n-1}$ and all previous stochastic measurement outcomes. The readout subsystem state is in a specific pure state, which can be ensured by the deterministic reset operation we describe shortly. Then, the current input $u_n$ is encoded in the quantum system via the parameterized quantum channel $\mathcal{U}(u_n)$, generating the state $\hat{\rho}_n^{\mathrm{MR,cond}} = \mathcal{U}(u_n)(\hat{\rho}_{n-1}^{\mathrm{M,cond}} \otimes |0\rangle \langle 0|^{\otimes R})$. In this work, $\mathcal{U}(u_n)$ takes the form of continuous evolution under Eq. (6) for a duration $\tau$, or the discrete gate-sequence $\hat{U}(u_n)$ depicted in Fig. 4a. The $R$ readout qubits are then measured per Eq. (2), and the observed outcome is represented as an $R$-bit string: $\mathbf{b}^{(s)}(n) = (b_{M+1}^{(s)}(n), \cdots, b_{M+R}^{(s)}(n))$. Here we consider simple 'computational basis' (i.e., $\hat{\sigma}^z$) measurements, where each bit simply denotes the observed qubit state. A given outcome $j$ occurs with conditional probability $\mathrm{Tr}(\hat{M}_j \hat{\rho}_n^{\mathrm{MR,cond}})$ as given by the Born rule, and the quantum state collapses to the new state $\hat{\rho}_n^{\mathrm{M,cond}} \otimes |\mathbf{b}_j\rangle \langle \mathbf{b}_j|$ associated with this outcome. Finally, all $R$ readout qubits are deterministically reset to the ground state (regardless of the measurement outcome); the quantum system is, therefore, in the state $\hat{\rho}_n^{\mathrm{M,cond}} \otimes |0\rangle \langle 0|^{\otimes R}$. This serves as the initial state into which the next input $u_{n+1}$ is encoded, and the above process is iterated until the entire input sequence $\mathbf{u}$ is processed. It is important to notice that $\hat{\rho}_n^{\mathrm{M}}$ depends on the observed outcome in step $n-1$, and thus, the quantum state and its dynamics for a specific shot are conditioned on the history of measurement outcomes $\{b_i^{(s)}(m)\}_{m<n}$.

By repeating the above process for $S$ shots, one obtains what is effectively a histogram of measurement outcomes at each time step $n$ as represented in Fig. 1. The output features are taken as the frequency of occurrence of each measurement outcome, as in ref. 9: $\bar{X}_j(n) = \frac{1}{S} \sum_{s=1}^{S} X_j^{(s)}(n; \mathbf{u})$, where $X_j^{(s)}(n; \mathbf{u}) = \delta(\mathbf{b}^{(s)}(n), \mathbf{b}_j)$ counts the occurrence of outcome $j$ at time step $n$. These features are stochastic unbiased estimators of the underlying quantum state probability amplitudes $x_j(n) = \mathbb{E}_{\mathcal{X}}[X_j^{(s)}(n; \mathbf{u})] = \lim_{S \to \infty} \bar{X}_j(n)$[9]. As noted in the main text, the final NISQRC output is obtained by applying a set of time-independent linear weights to approximate the target functional $\bar{y}_n = \mathbf{w} \cdot \bar{\mathbf{X}}(n)$. Importantly, during each shot $s \in [S]$, we execute a circuit with depth $N$; the total processing time is therefore $O(NS)$. If instead one re-encoded $N_m$ previous inputs prior to each successive measurement the processing time is $O(N_m NS)$: $N_m = O(N)$ if the entire past sequence is re-encoded as is conventionally done in QRC[26,27,50].

### The quantum Volterra theory and analysis of NISQRC

At any given time step $n$, the conditional dependence on previous measurement outcomes, presented in the "Methods" subsection "Generating features via conditional evolution and measurement", is usually referred to as *backaction*. Defining $\hat{\rho}_n^{\mathrm{MR}}$ as the effective pre-measurement state of the quantum system at time step $n$ of the

NISQRC framework, quantum state evolution from time step $n-1$ to $n$ can be written via the maps:

$$\hat{\rho}_n^{\text{MR}} = \mathcal{U}(u_n)\Big(\text{Tr}_{\text{R}}(\hat{\rho}_{n-1}^{\text{MR}}) \otimes |0\rangle\langle 0|^{\otimes R}\Big), \tag{8}$$

$$\hat{\rho}_n^{\text{M}} = \text{Tr}_{\text{R}}(\mathcal{U}(u_n)(\hat{\rho}_{n-1}^{\text{M}} \otimes |0\rangle\langle 0|^{\otimes R})) \equiv \mathcal{C}(u_n)\hat{\rho}_{n-1}^{\text{M}}, \tag{9}$$

which describes the reset of the post-measurement readout subsystem after time step $n-1$, followed by input encoding via $\mathcal{U}(u_n)$ into the full quantum system state. With an eye towards the construction of an I/O map, it proves useful to introduce the expansion of the relevant single-step maps $\mathcal{U}(u)$ and $\mathcal{C}(u)$ in the basis of input monomials $u^k$: $\mathcal{U}(u)\hat{\rho}^{\text{MR}} = \sum_{k=0}^{\infty} u^k \mathcal{R}_k \hat{\rho}^{\text{MR}}$ and $\mathcal{C}(u)\hat{\rho}^{\text{M}} = \sum_{k=0}^{\infty} u^k \mathcal{P}_k \hat{\rho}^{\text{M}}$. Then, via iterative application of Eq. (9), $\hat{\rho}_n^{\text{MR}}$ can be written as:

$$\hat{\rho}_n^{\text{MR}} = \sum_{k_1,\cdots,k_n=0}^{\infty} u_1^{k_1} \cdots u_n^{k_n} \mathcal{R}_{k_n}\Big(\mathcal{P}_{k_{n-1}} \cdots \mathcal{P}_{k_1}\hat{\rho}_0^{\text{M}} \otimes |0\rangle\langle 0|^{\otimes R}\Big). \tag{10}$$

The measured features $x_j(n)$ can then be obtained via $x_j(n) = \text{Tr}(\hat{M}_j \hat{\rho}_n^{\text{MR}})$.

In Supplementary Note 3, we show that these $x_j(n)$ obtained using the NISQRC framework can indeed be expressed as a Volterra series

$$x_j(n) = \sum_{k=0}^{\infty} \sum_{n_1=0}^{\infty} \cdots \sum_{n_k=n_{k-1}}^{\infty} h_k^{(j)}(n_1,\cdots,n_k) \prod_{\kappa=1}^{k} u_{n-n_\kappa} \tag{11}$$

in the infinite-shot limit. The existence of this manifestly time-invariant form is only possible due to the existence of an information steady-state, guaranteed for a quantum mechanical system under measurement.

Due to fading memory, the Volterra kernel $h_k^{(j)}(n_1,\cdots,n_k)$ characterizes the dependence of the systems' output at time $n$ on inputs at most $n_k$ steps in the past (recall $n_1 \le \cdots \le n_k$, see Eq. (11)). The evolution of $\hat{\rho}_n^{\text{MR}}$ up to step $n-n_k$, namely for all $i < n - n_k$, is thus determined entirely by the null-input superoperator $\mathcal{P}_0$. Then the existence of a Volterra series simply requires the existence of an asymptotic steady state for the memory subsystem, $\lim_{n\to\infty} \mathcal{P}_0^n \hat{\rho}_0^{\text{M}} = \hat{\rho}_{\text{FP}}^{\text{M}}$. As shown in the Supplementary Note 3, such a fixed point is usually ensured by the map $\mathcal{P}_0 \hat{\rho}^{\text{M}} = \mathcal{C}(0)\hat{\rho}^{\text{M}} = \text{Tr}_{\text{R}}(\mathcal{U}(0)(\hat{\rho}^{\text{M}} \otimes |0\rangle\langle 0|^{\otimes R}))$ being a CPTP map in generic quantum systems. This immediately indicates the fundamental importance of $\mathcal{P}_0$, the operator that corresponds to the single-step map of the memory subsystem under null input: it determines the ability of the NISQRC framework to evolve the quantum system to a unique statistical steady state, guaranteeing the asymptotic time-invariance property, and hence the existence of the Volterra series.

One byproduct of computing infinite-$S$ features $\{x_j(n)\}$ is that it enables us to approximately simulate $\{\bar{X}_j(n)\}$ in a very deep $N$-layer circuit for finite $S$, without sampling individual quantum trajectories under $N$ repeated projective measurements described in "Methods" subsection "Generating features via conditional evolution and measurement". In fact, given any $n$, once we evaluate a probability distribution $\{x_j(n) \ge 0\}$ satisfying $\sum_j x_j(n) = 1$, we can i.i.d. sample under this distribution vector for $S$ shots and construct the frequency $\{\tilde{X}_j(n)\}$ as an approximation of $\{\bar{X}_j(n)\}$. The validity of this approximation is ensured by the additive nature of loss functions in the time dimension. More specifically, given $Q$ input sequences $\{\mathbf{u}^{(q)} \in [-1,1]^N\}_{q\in[Q]}$, a general form of loss function is $\mathscr{L} = \frac{1}{QN}\sum_q\sum_n\mathcal{L}(\bar{\mathbf{X}}(n;\mathbf{u}^{(q)}))$. As shown in Appendix C5 of ref. 9, $\frac{1}{Q}\sum_q\mathcal{L}(\bar{\mathbf{X}}(n;\mathbf{u}^{(q)})) \approx \frac{1}{Q}\sum_q\mathcal{L}(\tilde{\mathbf{X}}(n;\mathbf{u}^{(q)}))$ in all orders of $\frac{1}{S}$-expansion for any $n \in [N]$, as long as $Q$ is large enough. This is because the probability distribution of $\{\tilde{X}_j(n)\}$ is exactly the same as the distribution (marginal in time slice) of $\{\bar{X}_j(n)\}$. Therefore, $\frac{1}{QN}\sum_q\sum_n\mathcal{L}(\tilde{\mathbf{X}}(n;\mathbf{u}^{(q)}))$ is a good approximation of $\mathscr{L}$.

In Supplementary Note 2 and Supplementary Note 3, we show that without the reset operation, the fixed-point memory subsystem density matrix is the identity, $\hat{\rho}_{\text{FP}}^{\text{MR}} = \hat{I}^{\otimes L}/2^L$. While this steady state is independent of the initial state and therefore possesses a fading memory, it can be shown that the I/O map it enables is entirely independent of all past inputs as well, so that Volterra kernels $h_k^{(j)} = 0$ for any $k \le 1$. This yields a trivial reservoir, unable to provide any response to its inputs $u$. Such single-step maps $\mathcal{C}(u)$ are referred to as *unital maps* (maps that map identity to identity) and must be avoided for the NISQRC architecture to approximate any nontrivial functional. The inclusion of reset serves this purpose handily, although we have found certain improper encodings with reset to still result in unital maps $\mathcal{C}(u)$ (e.g., setting $n_T = 1$ in the circuit ansatz depicted in Fig. 4(a)).

A more rigorous sufficient condition for obtaining a nontrivial functional map, referred to as a *fixed-point non-preserving* map in the main text, is that $\mathcal{C}(u)$ does not share the same fixed points for all $u$. It is equivalently $\mathcal{P}_k\hat{\rho}_{\text{FP}}^{\text{M}} \ne 0$ for some $k \ge 1$, due to the identity $\mathcal{C}(u)\hat{\rho}_{\text{FP}}^{\text{M}} = \hat{\rho}_{\text{FP}}^{\text{M}} + \sum_{k=1}^{\infty}u^k\mathcal{P}_k\hat{\rho}_{\text{FP}}^{\text{M}}$. We will prove the importance of these criteria in Supplementary Note 3. The breaking of this criteria will lead to a memoryless reservoir for all earlier input steps: if $\mathcal{P}_k\hat{\rho}_{\text{FP}}^{\text{M}} = 0$ for all $k \ge 1$, then $h_k^{(j)}(n_1,n_2,\cdots,n_k) \ne 0$ only if $n_1 = n_2 = \cdots = n_k = 0$. A similar result for quantum reservoirs characterized by quantum channels can also be found from Theorem 2 in ref. 53.

## Spectral theory of NISQRC: memory, measurement, and kernel structures

Recall that we can always define the spectral problem $\mathcal{P}_0\hat{\varrho}_\alpha^{\text{M}} = \lambda_\alpha\hat{\varrho}_\alpha^{\text{M}}$ where $\hat{\varrho}_\alpha^{\text{M}}$ are eigenvectors that exist in the $(2^M)^2 = 4^M$-dimensional space of memory subsystem states and whose eigenvalues satisfy $1 = \lambda_1 \ge |\lambda_2| \ge \cdots \ge |\lambda_{4^M}| \ge 0$. The importance of the spectrum of $\mathcal{P}_0$ is obvious from the definition of $\hat{\rho}_{\text{FP}}^{\text{M}}$ already. As $\hat{\rho}_{\text{FP}}^{\text{M}}$ is the fixed point of the map defined by $\mathcal{P}_0$, it must equal the eigenvector $\hat{\varrho}_1^{\text{M}}$ since $\lambda_1 = 1$. Then writing the initial density matrix in terms of these eigenvectors, $\hat{\rho}_0^{\text{M}} = \sum_\alpha d_{0\alpha}\hat{\varrho}_\alpha^{\text{M}}$, the fixed point becomes $\hat{\rho}_{\text{FP}}^{\text{M}} = \lim_{n\to\infty}\Big(\hat{\varrho}_1^{\text{M}} + \sum_{\alpha\ge2}d_\alpha^0\lambda_\alpha^n\hat{\varrho}_\alpha^{\text{M}}\Big)$. This not only reproduces the result $\lim_{n\to\infty}\mathcal{P}_0^n\hat{\rho}_0^{\text{M}} = \hat{\rho}_{\text{FP}}^{\text{M}}$ but also shows that the approach to the fixed point $\hat{\rho}_{\text{FP}}^{\text{M}} = \hat{\varrho}_1^{\text{M}}$ must be determined by the magnitude of $\lambda_2$; the smaller the magnitude, the faster terms for $\alpha \ge 2$ decay and hence, the shorter the memory time.

To see more directly how the spectrum of $\mathcal{P}_0$ influences the memory of inputs, it is sufficient to analyze the Volterra kernels in Eq. (1). Focusing on single-time contributions from $u_{n-p}$ to $x_j(n)$ at all orders of nonlinearity (multi-time contributions are exponentially suppressed, see Supplementary Note 4), these may be expressed as

$$\sum_{k=1}^{\infty} h_k^{(j)}(p^{\otimes k})u_{n-p}^k = \sum_{\alpha=2}^{4^M} \nu_\alpha^{(j)}\lambda_\alpha^{p-1}F_\alpha(u_{n-p}), \tag{12}$$

which can be viewed as a spectral representation of Volterra kernel contributions to the $j$th measured feature obtained via POVM $\hat{M}_j$. Here, $F_\alpha(u) = \sum_{k=1}^{\infty}c_{\alpha1}^{(k)}u^k$ define $4^M - 1$ *internal features*, so-called as they depend only on input encoding operators via $\mathcal{P}_k\hat{\varrho}_{\alpha'}^{\text{M}} = \sum_{\alpha=2}^{4^M}c_{\alpha\alpha'}^{(k)}\hat{\varrho}_\alpha^{\text{M}}$, and are in particular independent of the measurement scheme. Nontrivial $F_\alpha(u)$ and $c_{\alpha1}^{(k)}$ can be guaranteed if $\mathcal{P}_k\hat{\rho}_{\text{FP}}^{\text{M}} \ne 0$ for some $k \ge 1$. The dependence of observables on the measurement basis is via coefficients $\nu_\alpha^{(j)} = \text{Tr}(\hat{M}_j\mathcal{R}_0(\hat{\varrho}_\alpha^{\text{M}} \otimes |0\rangle\langle 0|^{\otimes R}))$. Crucially, the weighting of $F_\alpha(u_{n-p})$ for $p$ steps in the past is determined by eigenvalues $\lambda_\alpha^{p-1}$ of $\mathcal{P}_0$. For each $\alpha \ge 2$, it vanishes when we take a long time limit $p \to \infty$. This property is usually referred to as *fading memory*. It also clearly defines a set of distinct but calculable, memory fading rates $\{|\lambda_\alpha|\}_{\alpha\ge2}$.

Importantly, the ability to construct Volterra kernels and internal features enables us to approximately treat the infinite-dimensional function $x_j(n) = \mathcal{F}_j(u_{\le n})$ as a function with support only over a space

with *effective task dimension* $d_{eff} = O(n_M)$, representing $d_{eff}$ time steps in the past:

$$x_j(n) = \mathcal{F}_j(u_{\leq n}) \approx \mathcal{F}_j(u_{n-d_{eff}}, \cdots, u_{n-1}, u_n), \tag{13}$$

and we can interpret the fading memory function as a function: $y(n) \approx \mathcal{F}(u_{n-d_{eff}}, \cdots, u_{n-1}, u_n)$. In other words, at any given time NISQRC can approximate nonlinear functions that live in a domain of dimension $d_{eff}$.

### IBMQ implementation

We recall that the encoding circuit $\hat{U}(u_n) = \left(\mathcal{W}(J)\mathcal{R}_z(\boldsymbol{\theta}^z + \boldsymbol{\theta}^I u_n)\right.$ $\left.\mathcal{R}_x(\boldsymbol{\theta}^x)\right)^{n_T}$ for the experimental IBMQ implementation in the Results subsection "Experimental results on the quantum system" describes a composite set of single and two-qubit gates repeated $n_T$ times. Here $\mathcal{R}_{x,z}$ are composite Pauli-rotations applied qubit-wise, e.g., $\mathcal{R}_z = \otimes_i R_z(\theta_i^z + \theta_i^I u)$. $\mathcal{W}(J)$ defines composite two-qubit coupling gates, $\mathcal{W}(J) = \prod_{\langle i,i'\rangle}\mathcal{W}_{i,i'}(J) = \prod_{\langle i,i'\rangle} \exp\{-i(J\tau/n_T)\hat{\sigma}_i^z \hat{\sigma}_{i'}^z\}$ for neighboring qubits $i$ and $i'$ along a linear chain in the device and some fixed $J$. The rotation angles $\theta^{x,z,I}$ are randomly drawn from a positive uniform distribution with limits $[a, a+\delta]$, where $a = \frac{\tau}{n_T}\theta_{min}^{x,z,I}$ and $\delta = \frac{\tau}{n_T}\Delta\theta^{x,z,I}$. We find that letting the number of Trotterization steps $n_T = 3$ is sufficient to generate a well-behaved null-input CPTP map $\mathcal{P}_0$. Our hyperparameter choices are further tuned to ensure a memory time $n_M$ commensurate with the CE task dimension. The particular hyperparameter choices for the plot in Fig. 4 are $\theta_{min}^{x,z,I} = \{1.0, 0.5, 0.1\}$, $\Delta\theta^{x,z,I} = \theta_{min}^{x,z,I}$, $J = 1$, $n_T = 3$, and $\tau = 1$.

In the experiment, mid-circuit measurements and qubit resets are performed as separate operations, due to the differences in control flow paths between returning a result and the following qubit manipulation[55]. Related hardware complexities restrict us to a slightly shorter instance of the CE task than considered in Results' subsection "Practical machine learning using temporal data", with messages $m(n)$ of length $N = 20$, submitted in batches of 200 jobs with 100 circuits each and 125 observations (shots) per circuit in order to prevent memory buffer overflows. Regardless, using cross-validation techniques, we ensure that our observed training and testing performance is not influenced by limitations of dataset size. We also forego the initial washout period needed to reach $\rho_{FP}^{MR}$ for similar reasons. Finally, the $\mathcal{W}_{i,i'}(J)$ rotations in the two-qubit Hilbert space that implement $\mathcal{W}(J)$ are generated by the native echoed cross-resonance interaction of IBM backends[65], which provides higher fidelity than a digital decomposition in terms of CNOTs for Trotterized circuits[66].

## Data availability

The data generated for numerical results in this study have been deposited in the GitHub repository under the accession link https://github.com/skhanCC/NISQRC-Codes[67]. The raw experimental data obtained from *ibmq_algiers* are not available in the GitHub repository due to its huge size, and its access can be be made available to interested parties upon request. The processed experimental data are available at the Github repository. The data of experimental parameters in this study are provided in the Supplementary Note 9. No external data was used in this study.

## Code availability

The code used in this article is available in the GitHub repository https://github.com/skhanCC/NISQRC-Codes.

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

## Acknowledgements

The authors acknowledge the support from the ARO contract W911NF-19-C0092 (received by G.J.R.), DARPA contract HR00112190072 (received by H.E.T.), AFOSR award FA9550-20-1-0177 (received by H.E.T.), and AFOSR MURI award FA9550-22-1-0203 (received by H.E.T.). The views, opinions, and findings expressed are solely the authors' and not the U.S. government's. The authors acknowledge the use of IBM Quantum services for this work.

## Author contributions

F.H., S.A.K., G.A., and H.E.T. conceived the project. F.H. developed the theoretical model, and F.H. G.A. and S.A.K. performed the numerical simulations. G.E.R. and G.J.R. proposed the applications to channel equalization tasks and the circuit ansatz. S.A.K. and N.T.B. performed the experiments on the IBM Quantum platform and analyzed the data. H.E.T

supervised the project. F.H., S.A.K., G.A., and H.E.T. wrote the manuscript based on the contributions of all authors.

## Competing interests

The authors declare no competing interests.
