## [Peer Review File · Nature Communications]

Overcoming the Coherence Time Barrier in Quantum Machine Learning on Temporal DataREVIEWER COMMENTS

Reviewer #1 (Remarks to the Author):

In this article, Hu and collaborators develop a new scheme for quantum reservoir computing that allows them to perform learning on temporal tasks that require memory that is longer than the coherence time of the qubits in the system. The main contributions of their work involve the development of quantum Volterra series formalism for the guidance of the choice of the measurement basis, experimental implementation on the IBM cloud platform, and the discovery of the importance of the qubit reset after the measurement. These are important contributions to the field of quantum machine learning and I support the publication of this work in Nature Communications. My point-by-point questions and comments that could improve the readability of the article are listed below.

1. In the introduction, the authors claim that quantum systems could offer an enormous potential for scalable, energy-efficient and faster machine learning. Whereas there is some evidence for scalability and speed gain, energy efficiency seems less obvious. Could the authors argument more on this?
2. “The property that enables inference on an indefinitely-long input signal is intrinsic to the algorithm: it survives even in the presence of QSN, and does not require operating in a precisely-defined parameter subspace – and is thus unencumbered by barren plateaus.”
What is this property?
3. The authors implement learning on a classical task. Why this choice? Does it really make sense to perform classical tasks with a quantum neural network? Could the authors comment on their intuition why entanglement and quantum coherences should provide any advantage for classical tasks?
4. What do the colours inside the qubit circles in Figure 4(a) correspond to?
5. Figure 4(b) presents a comparison to logistic regression, but it is not specified what is trained with logistic regression. A classical reservoir? A perceptron?

6. Still in Figure 4(b), how do the authors explain the mismatch between experiments and simulation for connected vs unconnected, reset and no reset?

7. Still in Figure 4(b), why does a higher number of shots make the CE testing error rate higher for the split chain?

8. In methods I it is explained that in this particular quantum reservoir computing implementation, the authors present to the reservoir the whole input sequence for a single shot and then they start the whole sequence from scratch. This should be more clearly stated in the main text, as in other algorithms, the state probabilities are calculated for each input, so shots would be done multiple times for each input. Indeed, some parts of the methods I would be useful in the main text in order to understand the protocol.

9. The authors insist on the fact that their method strikes a balance between the input encoding steps and measurements – isn't this number the same? After each encoding gate the readout qubits are measured?

10. In the theory part the authors discuss the use of POVMs but it is not clear if and how they implement them experimentally on IBM computer, or do they implement simpler projective measurements.

11. In the supplementary material, above the equation A2 and in the remaining of the paper, the trace should be specified to be over M ?

12. Would it be possible to have a simulation result with the exact same configuration of the experiment to compare? It is hard to compare simulation and experiment because they have different numbers of qubits and connectivity.

13. What would be the scaling of the experiment time with the number of shots? Is it linear?

14. In Appendix F, how were parameters for h and f chosen?

15. In section I A, both transverse fields and Volterra kernels are called h , it doesn't seem like ideal choice.

16. The GitHub link for code doesn't work.

Reviewer #1 (Remarks on code availability):

The GitHub link for code doesn't work.

Reviewer #2 (Remarks to the Author):

(Quantum) reservoir computing is a machine learning framework for time series processing where an open (quantum) system is driven by an input time series and its response is post processed to create the transformed time series. For there to be any hope of practical quantum reservoir computing, the classical framework must be properly generalized. This puzzle has been missing many crucial pieces, locking the field into a proof-of-principle stage where assumptions can be unrealistic and even fatal flaws may go unnoticed. The main contributions of the manuscript at hand are exactly such missing pieces, and it should therefore be expected to steer the field as it matures. More specifically, they include generalizing Volterra series theory to account for measurement back-action in qubit-based systems, introducing the concept of persistent memory to quantum reservoir computing, and leveraging recent advances in partial mid-circuit measurements to introduce and experimentally test a scheme for NISQ reservoir computing that can tolerate sampling noise, measurement back-action and realistic coherence times. These are remarkable results. For example, the Authors use their quantum Volterra theory to show that some previously proposed schemes lack persistent memory which strongly limits their applicability whereas previous experiments have terminated at every measurement such that getting just one element of the desired output has the same scaling as getting the entire output time series with the introduced scheme. It is also worth pointing out that the Authors introduce a numerical method for simulating their scheme which is not constrained by circuit depth.

I think everyone working on quantum reservoir computing should read this paper. In any case, it is sure to be of considerable interest in the field.

The reported results seem correct and sound. The presentation is clear and conclusions are supported. The Methods together with the supplementary material cover the more technical aspects in great detail.

I have here listed a series of points aimed at further improving the manuscript.

1. It would be great if you could collect rules of thumb to design schemes with persistent memory into one or two paragraphs in some suitable place. I think this would be helpful for the field and complement your formal results. This could include for example the importance of the relative sizes of M and R , how to choose the input dependent part of the Hamiltonian (can local gates acting only on readout qubits be enough?) and so on.

2. I think you should briefly compare and contrast your scheme with that of Ref [31], which proposes real-time quantum reservoir computing in a multi-mode quantum optics platform. As far as I understand it also has persistent memory as it has separate modes for memory and readout, backaction is evaded and there is an implicit reset operation as the readout modes are replaced at every time-step.

3. If possible, can you briefly comment in, e.g., Discussion on the prospects of a quantum Volterra theory suitable for qudits or continuous-variable systems?

4. Minor point. I noticed that the text features both “backaction” and “back-action”. You should choose one.

I recommend the publication of this work in Nature Communications after the Authors have addressed the points.

Reviewer #2 (Remarks on code availability):

The GitHub profile <https://github.com/skhanCC> exists but I see no repository called NISQRC-Codes there.

The first time I heard that the code is supposedly available is here in this review form. I did not spot any mention of this in the main or supplementary article files.

Reviewer 1

The Reviewer's original comments are in black, our responses are in red, and changes to the manuscript are shown in blue. All sections, equations, figures, tables which are referred in response are all indexed with indices in the newly resubmitted manuscript.

In this article, Hu and collaborators develop a new scheme for quantum reservoir computing that allows them to perform learning on temporal tasks that require memory that is longer than the coherence time of the qubits in the system. The main contributions of their work involve the development of quantum Volterra series formalism for the guidance of the choice of the measurement basis, experimental implementation on the IBM cloud platform, and the discovery of the importance of the qubit reset after the measurement. These are important contributions to the field of quantum machine learning and I support the publication of this work in Nature Communications. My point-by-point questions and comments that could improve the readability of the article are listed below.

1. In the introduction, the authors claim that quantum systems could offer an enormous potential for scalable, energy-efficient and faster machine learning. Whereas there is some evidence for scalability and speed gain, energy efficiency seems less obvious. Could the authors argue more on this?

We expect an energy gain to present itself when quantum systems are used to perform machine learning on weak noisy signals (e.g. classification of light waves emanating from a weakly illuminated scene). For computation using a classical ML algorithm running on a digital computer, such signals would need to undergo amplification and digitizing, and then processing, with the additional need of memory depending on the computation scheme. Each such step expends energy. In contrast, a quantum system can directly interface with such signals, especially weak signals, without the need for these intermediate steps and directly implementing information processing in an analog fashion. The experiment we conduct (channel equalization on the IBM device) is one instance of this, albeit somewhat contrived: The messages are encoded on microwave signals generated by an AWG, that are then attenuated and brought down to a few-photon level when they act on the qubits. The whole information encoding here is essentially happening at a single-photon level (e.g. rotations by π correspond to a single photon exchange between the microwave signal and an individual qubit). While we have not carried out a direct comparison of the energy cost in using a classical ML algorithm running on a digital computer against a quantum system, we believe the aforementioned factors could enable an energetic advantage for information processing using quantum systems, especially when coupled with scalability advantages.

2. “The property that enables inference on an indefinitely-long input signal is intrinsic to the algorithm: it survives even in the presence of QSN, and does not require operating in a precisely-defined parameter subspace – and is thus unencumbered by barren plateaus.” What is this property?

This property is that of avoiding measurement-induced thermalization at long times under repeated measurements, enabled in the NISQRC algorithm by injection of purity into the quantum state via a deterministic reset scheme. Measurement-induced thermalization prevents a persistent, finite memory time in the steady-state, and our algorithm is able to overcome it. We have amended the above sentence to provide some clarity:

“The property that enables inference on an indefinitely-long input signal - the ability to avoid measurement-induced thermalization at long times under repeated measurements due to a deterministic reset protocol - is intrinsic to the algorithm: it survives even in the presence of QSN, and does not require operating in a precisely-defined parameter subspace – and is thus unencumbered by barren plateaus.”

3. The authors implement learning on a classical task. Why this choice? Does it really make sense to perform classical tasks with a quantum neural network? Could the authors comment on their intuition why entanglement and quantum coherences should provide any advantage for classical tasks?

We agree with the Reviewer that there seems to be a consensus in the QML community that quantum systems do not offer an advantage in ML tasks on ‘classical data’. One of the reasons for this belief stems from the additional resources needed to ‘quantize’ the classical data. Our point of view is that such resource costs can be overcome for ML tasks on weak electromagnetic signals. For instance, quantum neural networks implemented in circuit QED can interface directly with weak RF signals, enabling potential advantages in their processing as compared to collecting, amplifying and digitizing these signals before processing them in classical digital electronics (please see also Comment #1). In addition to this rationale for the chosen task, the channel equalization task is also a popular choice for benchmarking reservoir computing algorithms (please see also Comment #14).

Secondly, the question of whether quantum resources such as entanglement in particular can provide advantages for classical information processing is an open one, and one we hope our framework can help address by enabling complex temporal information processing tasks to be performed using quantum systems. For some first hints in this direction, in this work we have for example shown how performance of the channel equalization task - a benchmark machine learning task on temporal data - improves with increasing qubit connectivity (while retaining the same total number of qubits, Fig. 3). Our analysis shows that increasing connectivity increases the Jacobian rank of the input-output map, which in simple terms enables the realization of more

linearly-independent functionals (for more details, see Appendix E). The increase in connectivity is one crude measure of the degree of entanglement of the quantum system; our NISQRC algorithm provides the means to study this dependence further in larger systems and more complex tasks, to quantitatively identify whether quantum advantages can exist for classical machine learning tasks on weak signals. A previous paper (Ref. [9] in the revised manuscript) also provides an analysis showing that correlations in a finitely sampled quantum system can increase the function expression capacity on classical input data (data here was static).

4. What do the colours inside the qubit circles in Figure 4(a) correspond to?

We thank the Reviewer for bringing this to our attention. The filled colors in the schematic, which is directly obtained from the IBM Quantum website for *ibmq_algiers* (https://quantum.ibm.com/services/resources?tab=systems&type=Falcon&system=ibmq_algiers), are meant to indicate some chosen qubit parameter for the device. The specific schematic in the previous version of the manuscript depicted 'Readout assignment error'. We have now updated the schematic so that the filled colors represent the qubit T_1 times, which is a more relevant quantity for our work, and have also provided the colorbar that quantifies these T_1 times. Note that the T_1 times are shown for one specific experimental run - the split chain - since the lifetimes vary depending on the run, as shown in Tables I through III in the SI.

5. Figure 4(b) presents a comparison to logistic regression, but it is not specified what is trained with logistic regression. A classical reservoir? A perceptron?

Direct logistic regression in Figure 3(a) and 4(b) refers to a simple logistic model applied directly to the current input. This is a single layer perceptron with a softmax activation function trained by minimizing cross-entropy loss. In the original manuscript this was described in the last paragraph of SI F. We have added further details to SI F, and have also added additional clarification to the main text when Fig 3 is discussed. Finally, we now provide explicit references to SI F for details about logistic regression, both in text and in the figure caption.

6. Still in Figure 4(b), how do the authors explain the mismatch between experiments and simulation for connected vs unconnected, reset and no reset?

We believe there are three possible reasons for the mismatch. Firstly, our simulations assume a simplified noise model, with a distribution of coherence times for the qubits, and furthermore assuming no gate or readout errors. The experimental device in practice of course experiences such errors. Secondly, due to the complexity of implementing mid-circuit measurements, a relatively recent addition to IBM Quantum, as well as resource limitations, the total required shots for a given input message cannot all be obtained in sequence, but are obtained in small batches, sometimes spread out over several hours. As a result, the IBM Quantum device can exhibit significant parameter drifts from one type of experiment to another (e.g. from connected chain

with reset to the split chain with reset), and even during the course of collecting all shots for a given type of experiment. Such parameter drifts - clearly seen from device parameters indicated for the three experiment types in Tables I through III - can lead to a difference in the simulated device and the actual experiment, which could lead to difference in the observed performance in either direction.

Finally, we also noticed a systematic change in device coherence times for the first experiment (connected chain with reset) in comparison to the other two; this is also documented in Tables I through III. We see that our experiment on the connected chain with reset has the highest coherence lifetimes. On the other hand, the experiments on the split chain with reset, and the connected chain without reset, both exhibit lower coherence times. In particular, some qubits in these experiments even have coherence times that are approaching the memory time needed for our particular realization of the channel equalization task, which can start to hinder performance. We believe these reasons could explain the difference between the simulation and experiments in these two cases.

We have summarized the above discussion in the main text, Sec. I D. We thank the Reviewer for helping us clarify our experimental results.

7. Still in Figure 4(b), why does a higher number of shots make the CE testing error rate higher for the split chain?

As mentioned in the response to the previous comment, we believe this is again primarily because not all shots even for a given experiment are obtained in sequence, and were sometimes collected from experimental runs separated by several hours. Due to this, some amount of variation in performance with increase in shots can be attributed to device parameter drifts.

Secondly, one can note that there is sometimes a (much smaller) non-monotonicity even in the simulated device results, which obviously exhibits no device drifts. This is explained by our restriction to a limited number of input messages, again due to the technical resource reasons above. This greatly limited both our training and testing set sizes for any given number of shots, which can lead to variance in testing set performance. We performed cross-validation over different permutations of the training and testing set to alleviate this effect, but some signatures of it still remain.

Again, we thank the Reviewer for helping us clarify these results; we have also summarized the above in the discussion of our results in Sec. I D.

8. In methods I it is explained that in this particular quantum reservoir computing implementation, the authors present to the reservoir the whole input sequence for a single shot and then they start the whole sequence from scratch. This should be more clearly stated in the main text, as in other algorithms, the state probabilities are calculated for each input, so shots would be done multiple

times for each input. Indeed, some parts of the methods I would be useful in the main text in order to understand the protocol.

We thank the Reviewer for this suggestion, and have now indeed moved some of the content from Methods I to the two paragraphs proceeding Eq (4). We have reworked this section from the paragraph beginning “We begin by providing a fundamental description of both the information input and output processes that enable general time series processing with quantum systems, before specializing to the NISQRC algorithm..” to the line “This complete architecture, from the quantum circuit generating measurement outcomes for a given input, to the construction of weighted output features, is depicted schematically in Fig. 1” to provide at an early point in the main text a clear description of both the NISQ algorithm as implemented in this paper and its fundamental constituents so that these tools can be generalized.

As the Reviewer correctly stated, we perform S shots for each message presented to the NISQRC algorithm, where a message consists of N sequential inputs which pass through the quantum system in the iterative evolve-measure-reset process. Following a detailed description of this process for a single shot, we now clarify:

“The above description yields a set of N measurement outcomes $\{b^{(s)}(n)\}$ observed in a single shot s of the quantum circuit. In order to obtain statistics and therefore output features as expected values of observables M_j , we perform repetitions of this circuit for a given u : the total execution time is NS , linear with respect to shots S and input length N .”

9. The authors insist on the fact that their method strikes a balance between the input encoding steps and measurements – isn't this number the same? After each encoding gate the readout qubits are measured?

We thank the Reviewer for pointing out this potential source of confusion, which we have now improved. The original sentence in the abstract was not intended to imply that we balance the “number of the input encoding steps” and “number of measurements” (as the Reviewer observes, these two numbers are the same). What we meant was that in our proposed scheme, repeated measurements can be performed such that the quantum system (and hence outputs extracted from it) retain memory of correlations in the streaming data; more precisely, these correlations are not ‘forgotten’ due to the action of said measurements. This enables the input data to be encoded in streaming fashion, without having to restart the entire circuit, thereby minimizing circuit executions. We have modified the phrasing in the abstract to better convey this message:

“NISQRC exploits mid-circuit measurements with deterministic reset operations, which reduces the number of circuit executions but still endows the quantum system with an appropriate-length persistent temporal memory to capture the desired time-domain correlations in streaming data.”

10. In the theory part the authors discuss the use of POVMs but it is not clear if and how they implement them experimentally on IBM computer, or do they implement simpler projective measurements.

While our framework allows for general POVMs, for experiments on IBM Quantum we are restricted to projective measurements. We have now emphasized this in the main text after Eq. (2) more explicitly,

“Here we will consider a simple measurement in the readout qubit computational basis, described by...”

The language of POVMs is used to help describe the overall measure-reset operation:

“The resulting measure-reset operation we employ throughout this paper is formally described by the POVM operators $E_j = K_j^\dagger K_j$ in Eq. (2), with non-hermitian Kraus operators $K_j = |b_0\rangle\langle b_j|$ ”

11. In the supplementary material, above the equation A2 and in the remaining of the paper, the trace should be specified to be over M?

In this paper, we have defined operators \hat{M}_j and $\hat{\rho}^{MR}_n$ to be the quantum operators on the *composite system* (see Eq.(2)), and not just operators on either the Memory or Readout subsystem; more precisely, both are operators defined on the joint $L=M+R$ qubit Hilbert space. Therefore, when we write down the expression $\text{Tr}(\hat{M}_j \hat{\rho}^{MR}_n)$, we must perform a trace over the composite Hilbert space, defined by the notation “Tr” unambiguously.

12. Would it be possible to have a simulation result with the exact same configuration of the experiment to compare? It is hard to compare simulation and experiment because they have different numbers of qubits and connectivity.

For our experimental results (solid lines in Fig. 4), which were performed on IBM Quantum, our accompanying simulations (dashed lines in Fig. 4) always consider the exact same configuration as the experiment, namely a linear chain of 7 qubits with nearest neighbor connectivity. Hence our simulation results can be compared directly with the experiments on IBM Quantum, for all three cases shown in Fig. 4.

Perhaps the confusion has arisen due to Fig. 3, which shows a different connectivity than the experiments. However, we emphasize that Fig. 3 describes a completely different quantum system, namely a (continuous-time) quantum annealer, and hence should not be directly compared to the experiments on IBM Quantum parameterized circuits in Fig. 4. To clarify the difference, we have amended the discussion of the results of Fig. 3:

“We now perform the CE task using the NISQRC algorithm on a simulated (2 + 4)-qubit reservoir under the ansatz of Eq. (5), namely describing a quantum annealer (see Fig. 3). We will later demonstrate the same task in experiments using a completely different quantum system: parameterized quantum circuits using IBM Quantum (see Fig. 4).”

13. What would be the scaling of the experiment time with the number of shots? Is it linear?

The experiment runtime scales linearly with the number of shots. The additions to Sec. 1A in response to question 8 state this directly, and we also emphasize this point in the penultimate paragraph of Section IB:

“As shown in schematic Fig.1, incorporating partial measurement with reset in NISQRC does not require such a re-encoding; the entire input sequence can be processed in any given measurement shot S , enabling online processing with an $O(NS)$ runtime, while maintaining a controllable memory timescale.”

14. In Appendix F, how were parameters for h and f chosen?

This channel equalization task was presented in many reservoir computing papers, for example

- <https://www.science.org/doi/10.1126/science.1091277>
- <https://www.nature.com/articles/srep00287>
- <https://www.nature.com/articles/srep14945>

We chose those parameters of “ h ” and “ f ” from the papers above. The only difference is that we removed the interference from the two-step future symbols, and also enhanced the nonlinear coefficients in “ f ” a little bit to make “ f ” non-invertible. We now cite the first paper above in SI F, and add an explanation on how we choose the parameters.

15. In section I A, both transverse fields and Volterra kernels are called h , it doesn't seem like ideal choice.

We thank the Reviewer for pointing out this source of possible confusion; we have now changed the notation of single qubit drives/fields from “ h ” to “ η ” in the manuscript.

16. The GitHub link for code doesn't work.

The Github repo containing the relevant codes was unfortunately set to ‘Private’; it has now been made ‘Public’ for access by external users, and can be viewed at the original link, for convenience reproduced here.

Now we also added a section of “Code Availability” after “Methods”.

Remarks on code availability:

The GitHub link for code doesn't work.

(See previous response)

Reviewer 2

The Reviewer's original comments are in black, our responses are in red, and changes to the manuscript are shown in blue. All sections, equations, figures, tables which are referred in response are all indexed with indices in the newly resubmitted manuscript.

(Quantum) reservoir computing is a machine learning framework for time series processing where an open (quantum) system is driven by an input time series and its response is post processed to create the transformed time series. For there to be any hope of practical quantum reservoir computing, the classical framework must be properly generalized. This puzzle has been missing many crucial pieces, locking the field into a proof-of-principle stage where assumptions can be unrealistic and even fatal flaws may go unnoticed. The main contributions of the manuscript at hand are exactly such missing pieces, and it should therefore be expected to steer the field as it matures. More specifically, they include generalizing Volterra series theory to account for measurement back-action in qubit-based systems, introducing the concept of persistent memory to quantum reservoir computing, and leveraging recent advances in partial mid-circuit measurements to introduce and experimentally test a scheme for NISQ reservoir computing that can tolerate sampling noise, measurement back-action and realistic coherence times. These are remarkable results. For example, the Authors use their quantum Volterra theory to show that some previously proposed schemes lack persistent memory which strongly limits their applicability whereas previous experiments have terminated at every measurement such that getting just one element of the desired output has the same scaling as getting the entire output time series with the introduced scheme. It is also worth pointing out that the Authors introduce a numerical method for simulating their scheme which is not constrained by circuit depth.

I think everyone working on quantum reservoir computing should read this paper. In any case, it is sure to be of considerable interest in the field.

The reported results seem correct and sound. The presentation is clear and conclusions are supported. The Methods together with the supplementary material cover the more technical aspects in great detail.

I have here listed a series of points aimed at further improving the manuscript.

1. It would be great if you could collect rules of thumb to design schemes with persistent memory into one or two paragraphs in some suitable place. I think this would be helpful for the field and complement your formal results. This could include for example the importance of the relative sizes of M and R , how to choose the input dependent part of the Hamiltonian (can local gates acting only on readout qubits be enough?) and so on.

We thank the Reviewer for this suggestion; we have now included a brief discussion of the insights provided by our analysis on the design of qubit-based systems for time-series processing, in the ‘Discussion’ section. This includes the essential component of deterministic reset post-measurement, some heuristics on the relative size of M and R , as well as the ability to control the memory time using the unmonitored portion of the dynamics:

“Finally, by characterizing the general input-output map, we believe the QVT can prove invaluable in identifying general design principles for qubit-based systems used for online processing, even beyond this crucial reset component. For example, while measuring some fraction of qubits is essential for extracting information, measuring all qubits imposes a trivial memory time. We employ $M \sim R$ in this work, but an optimal separation of memory and readout qubits may depend on the specific task. A simple rule of thumb is to choose M/R , together with τ and other drive strengths, to match the memory time of the physical system to the longest correlation time in the data. In the Channel Equalization task studied in Section IC, the data correlation time is fixed by the choice of the distorting channel (see Appendix F) and we have then chosen M/R to endow the quantum system with a memory time – calculated through the QVT formalism – that matches that time, about 8 steps (recall memory time is measured in number of encoding steps). Especially, the duration of the unmonitored dynamics, τ , has been chosen to be long enough to generate non-linear kernels that match the known order of the non-linearity of the distorting channel, but short enough to avoid limiting the memory time by the shortest T_1 . Here Fig. 2c, calculated through QVT, acts as a very helpful guide. We have observed that even when there is a large spread in T_1 , the physical memory time may be longer than the shortest T_1 , presumably through the delocalization of the information on longer-lived memory qubits in the circuit. Finally, qubit connectivity, analyzed in Figure 3, can help with the generation of functions that are sufficiently complex to match the functional complexity of the task. We have not carried out here an exhaustive study of the design principles for more complex or general classes of tasks, but we hope that the simple guidelines we have followed for designing an experimental reservoir to accurately carry out equalization on RF-encoded messages illustrates the utility of QVT in the design of a hardware reservoir.”

2. I think you should briefly compare and contrast your scheme with that of Ref [31], which proposes real-time quantum reservoir computing in a multi-mode quantum optics platform. As far as I understand it also has persistent memory as it has separate modes for memory and readout, backaction is evaded and there is an implicit reset operation as the readout modes are replaced at every time-step.

We have carefully read Ref. [31], which provides and analyzes a proposal for scalable quantum reservoir computing using optical modes. As mentioned by the Reviewer, there are several components that at first glance have parallels between the two approaches, however we are unable to say anything definitive about the comparison of the two approaches without a detailed analysis. The key distinction of Ref. [31] is the extraction of information from *bosonic* quantum modes in Gaussian states using Gaussian (e.g. homodyne) measurements. In general, such

measurements have different backaction characteristics in comparison to the projective measurements on qubit-based systems that we primarily analyze in our work.

Interestingly, another article by the same group, Ref. [51], is even more relevant in this context. In fact, in Ref. [31] the authors say *“Furthermore, the act of measuring yields backaction into the system that may also negatively affect the performance of the reservoir. Two strategies to overcome this last issue in qubit platforms have been recently proposed in Ref. [28] (Ref. [51] in our manuscript) using weak measurements and partial sequence repetition (rewinding)”*. The essential components of our NISQRC scheme - partial readout and deterministic reset - are proposed to allow us to overcome the role of measurement backaction *without* using weak measurements and *without* having to repeat parts of the input sequence.

We have added the following sentences across the manuscript to provide some comparisons between our work and related prior work:

We note that an alternative scheme to remedy this issue has been suggested in Ref. [51], which relies on information extraction through continuous weak measurement.

and in the Discussion section:

We note here that continuous weak measurements, analyzed in Ref. [51], provides an alternative approach to endowing the reservoir with a finite persistent memory and can be analyzed with QVT for its task-specific optimization.

3. If possible, can you briefly comment in, e.g., Discussion on the prospects of a quantum Volterra theory suitable for qudits or continuous-variable systems?

The development of the quantum Volterra theory in this work relies simply upon the linearity of unmonitored evolution of the density matrix describing the state of a general quantum system, together with the Kraus operator description of quantum measurement. Both these facts also apply equally to qudits or continuous variable (CV) quantum systems. Furthermore, we have not been required to call upon the Hilbert space dimension of the quantum system (or subsystems) in our derivation. As a result, the formalism naturally encompasses qudits as well as CV systems.

However, the ability to practically compute quantum Volterra kernels will be affected by Hilbert space dimension. In particular, while the formalism would apply directly to CV systems represented using density matrices in a truncated Hilbert space spanned by Fock states, such a representation is clearly not the most efficient one for CV systems, where alternative representations (such as a coherent state bases) may be better. The development of a more tailored quantum Volterra theory for more general systems is an open problem.

We have summarized the above in the ‘Discussion’ section:

While we have applied the QVT to qubit-based circuits, our analysis does not make an explicit assumption on the Hilbert space dimension of the quantum system, and allows for completely general measurements through its formulation in terms of POVMs; as a result, it can be applied to other finite-level quantum systems such as qudits [59], and can be extended to continuous-variable quantum systems [28,31]. We therefore believe the QVT provides the ideal framework to analyze the memory and computational capacity of temporal information processing schemes using general quantum systems and their associated measurement protocols.

4. Minor point. I noticed that the text features both “backaction” and “back-action”. You should choose one.

We have now changed all instances of “back-action” to “backaction”.

I recommend the publication of this work in Nature Communications after the Authors have addressed the points.

Remarks on code availability:

The GitHub profile <https://github.com/skhanCC> exists but I see no repository called NISQRC-Codes there.

The first time I heard that the code is supposedly available is here in this review form. I did not spot any mention of this in the main or supplementary article files.

The Github repo containing the relevant codes was unfortunately set to ‘Private’; it has now been made ‘Public’ for access by external users, and can be viewed at the original link, for convenience reproduced here. We have also added a section after Methods to indicate the availability of the code.

REVIEWERS' COMMENTS

Reviewer #1 (Remarks to the Author):

The authors have replied to all my comments and I support the publication of their manuscript. A single remaining comment I have concerns my first question about the energy efficiency. I think the authors' answer is quite speculative - the quantum reservoir's signal also needs to be amplified before the measurement, and the measurement also has some energy cost just as digitization. I would be more careful with this energy efficiency statement.

Reviewer #1 (Remarks on code availability):

The code is working and can be run.

Reviewer #2 (Remarks to the Author):

Thank you for taking the time to address my points. Everything is in order and I have no further suggestions to make.

I recommend the work is published without further delay.

Reviewer #2 (Remarks on code availability):

While I did not run the code I took a look and have no reason to doubt it does its job and can be easily used by anyone proficient in Python.

Reviewer 1

The authors have replied to all my comments and I support the publication of their manuscript. A single remaining comment. I have concerns my first question about the energy efficiency. I think the authors' answer is quite speculative - the quantum reservoir's signal also needs to be amplified before the measurement, and the measurement also has some energy cost just as digitization. I would be more careful with this energy efficiency statement.

A: We replace the phrasing “energy-efficient” with “resource-efficient”. The latter has some concrete evidence. For example, Ref. [21] by Huang et al presented that “quantum machines could learn from exponentially fewer experiments than conventional experiments”.

Remarks on code availability:

The code is working and can be run.